



# Application of geophysical tools for tree root studies in forest ecosystems in complex soils

Ulises Rodríguez-Robles[1], Tulio Arredondo[1], Elisabeth Huber-Sannwald[1], José Alfredo Ramos-Leal[2], Enrico A. Yépez[3]

[1]División de Ciencias Ambientales, Instituto Potosino de Investigación Científica y Tecnológica, Camino a la Presa de San José # 2055, Lomas 4ta Sección, C.P. 78216 San Luis Potosí, S.L.P., México.
[2]División de Geociencias Aplicadas, Instituto Potosino de Investigación Científica y Tecnológica, Camino a la Presa de San José # 2055, Lomas 4ta Sección, C.P. 78216 San Luis Potosí, S.L.P., México.
[3]Departamento de Ciencias del Agua y Medio Ambiente, Instituto Tecnológico de Sonora, 5 de Febrero 818 Sur, Col. Centro, C.P. 8500 Ciudad Obregón, México.

*Correspondence to*: J. Tulio Arredondo (tulio@ipicyt.edu.mx)

**Abstract.** While semiarid forests frequently colonize rocky substrates, knowledge is scarce on how roots garner resources in these extreme habitats. The Sierra San Miguelito Volcanic Complex in Central Mexico exhibits shallow soils and impermeable rhyolitic-rock outcrops, which impede water movement and root placement beyond the soil matrix. However, rock fractures, exfoliations, and soil pockets potentially permit downward percolation and root growth. With ground penetrating radar (GPR) and electrical resistivity tomography (ERT), two geophysical methods advocated by Jayawickreme et al. 2014 to advance root ecology, we studied root and water distribution in shallow-rocky-soils and rock fractures in a semiarid forest. We calibrated geophysical images with in-situ root measurements, and then extrapolated root distribution over larger areas. With GPR, we identified fine and coarse pine and oak roots with 6 to 75 mm diameters at differential depths in soil and fractures; besides, trees anchored their trunks with coarse roots underneath rock outcroppings. With ETR, we tracked monthly changes in humidity at the soil/bedrock interface, which clearly explained spatial root distribution of both tree species. Geophysical methods have enormous potential in elucidating root ecology. More interdisciplinary research could advance our understanding in belowground ecological niche functions and their role in forest ecohydrology and productivity.

## 1 Introduction

Strategies of plant water use and mechanisms of water transport at the soil–plant–atmosphere continuum are critical to understand ecosystem functioning in arid and semiarid regions, where plant productivity is primarily limited by water availability (Prieto et al., 2012;Burgess and Bleby, 2006;Li et al., 2007). Roots' major functions are absorbing water and mineral nutrients, as well as supporting stems and anchoring plants to the ground (Prieto et al., 2012). Growing roots change soil structure, displace pore water and gas, and increase porosity (Jackson et al., 1996). Plant water balance and physiological processes depend on the control of root water uptake (Anderegg and HilleRisLambers, 2016). Placement of



roots at different soil depths favors both spatial and temporal resource partitioning and potentially effective resource exploitation of whole soil profiles, thereby enhancing biomass production (Fernandez et al., 2000;Brooks et al., 2002;Hultine et al., 2003;Renee et al., 2010). Many ecosystems with shallow soils (<1 m) are located in water-limited climatic regions with highly variable, seasonal precipitation, where a minimum water storage potential by the substrate is paramount to

maintain perennial vegetation cover (Rose et al., 2003;Schwinning, 2010;Rodriguez-Robles et al., 2015). There exist a fair number of studies on hydrological aspects of plants from semiarid regions, in particular, from sites where vertical root development is not restricted by hardened soil layers such as in karstic regions (Schwinning and Ehleringer, 2001;Poulos et al., 2007;Lebourgeois et al., 1998;Estrada-Medina et al., 2013). Few studies, however have examined semiarid forest ecosystems with shallow soils forming over bedrock, cemented horizons, or strongly developed argillic horizons that impede

downward water movement and root growth (Andrews et al., 2005;Katra et al., 2008;Rodriguez-Robles et al., 2015).

In semiarid climates, poorly developed shallow soils over water-impermeable substrates rarely exhibit sufficient water storage capacity to maintain forest ecosystems. Empirical evidence suggests that trees and shrubs growing on this type of substrate are able to access water from weathered bedrock once water supply from the top soil becomes exhausted (Querejeta et al., 2007;Schwinning, 2008). Still, the geological context of shallow soils in plant-water relations is controversial both in

terms of the physical source of water and the adaptive mechanisms to thrive under these limiting water conditions. This has been the focus of recent ecohydrological studies (Tokumoto et al., 2014;Schwinning, 2013;Rodriguez-Robles et al., 2015;Estrada-Medina et al., 2013). Because of methodological difficulties, the impracticality of bedrock excavation and a general lack of specific research tools to study root distribution *in-situ*, little is known about species-specific rooting patterns and growth strategies of forest ecosystems colonizing shallow soils over bedrock. Complementary methods are needed to

simultaneously study vertical root distribution and seasonal soil humidity patterns to elucidate potentially diverse species-specific adaptations to complex geoecohydrological conditions.

With surface geophysical methods, such as electrical resistivity tomography (ERT), it is possible to monitor water content at soil-bedrock depths between 2.5 and 17 m and at frequent time intervals (Beff et al., 2013). ERT is a nondestructive, geoelectrical method to examine soil properties (Martinez-Pagan et al., 2013); it allows the generation of

two- and/or three-dimensional images and maps depicting both the spatial and temporal variation in soil electrical conductivity, corresponding to variations in soil water content (Cosentini et al., 2012), and singularities like cracks and fractures (Travelletti et al., 2012). The resistivity of rocks and soil may vary depending on their water content, water salinity and mode of pore distribution, with a wide range of values (1-$10^9$ $\Omega \cdot m^{-1}$); low values indicate high water content and high values low water content (Orellana and Silva, 1982). ERT has traditionally been used in geological prospecting (Chrétien et

al., 2014;Sudha et al., 2009;Wang et al., 1991), but is now also frequently applied in hydrological, agricultural and environmental studies (Srayeddin and Doussan, 2009;Jackson et al., 2000).

Ground-penetrating radar (GPR) is an effective and rapid tool for geophysical surveys because it is nondestructive and delivers real-time information (Parsekian et al., 2015). The GPR permits the use of a variety of antennas with different (high



and low resolution) frequencies for the examination of different substrates and to be used by multiple disciplines. The application of GPR ranges from characterizing subsurface stratigraphy (Adepelumi and Fayemi, 2012) and spatial extent of weathered blocks and fracture-cracked systems (Ogretmen and Seren, 2014) to measurements of soil water content (da Silva et al., 2004) and the determination of belowground tree root diameters (with minimum diameter of 5 mm) in forest and urban settings (Tanikawa et al., 2013;Ow and Sim, 2012;Hruska et al., 1999). A combined application of GPR with ERT explored the distinction and distribution of roots with different diameters over a broad range of soil conditions in south-eastern United States (Butnor et al., 2001). According to their study, soils with high electrical resistivity are the most amenable for root detection with GPR. More recent studies (Zhu et al., 2014;Borden et al., 2014) have tried to track the direction (vertical or horizontal) of root growth and to evaluate the efficiency of GPR in mapping coarse root systems and estimating root biomass under field conditions. However, the successful application of GPR for root detection depended on specific site characteristics, as numerous factors (e.g., soil physico-chemical properties, water content, terrain conditions, etc.) may interfere with signal transmission and thus the resolution of root axes (Table 1).

To explore the potential of these geophysical methods in ecology, we examined ecohydrological processes at the soil-bedrock-plant root interface in a mixed forest ecosystem in the mountainous region of Sierra San Miguelito (SSM) situated at the transition of the arid desert scrub biome in the North and the semiarid grassland biome in the South of the Mexican Central Plateau. Since the presence and expansion of pine-oak forests in this macroclimatic semiarid region cannot be explained by mere climate conditions, we used geophysical methods to elucidate the geological and edaphic conditions as well as potential root adaptations, to help explain the ecohydrological functioning of this azonal forest. The SSM is a volcanic complex of impermeable rhyolitic rocks, whose surface layers have been highly weathered by exfoliation processes (peeling off in sheets). The forest ecosystems are characterized by shallow, poorly developed soils with high litter and organic matter content (<25 cm deep) (Perez et al., 2009). Recent studies suggested that native tree species may be able to extract water directly from subsurface bedrock (Schwinning, 2013;Proust et al., 2011;Tokumoto et al., 2014), however, most of these studies have focused on water-permeable rock types (e.g., limestones). Rodriguez-Robles et al. (2015) suggested that specialized root systems of tree colonizing the shallow rocky soils of SSM explore large regolith rocky areas and thereby increase the likelihood of finding stored water in cracks. However, it is unclear if the water supply encountered in cracks fulfills the water demand of two coexisting tree species especially during frequently recurring extended seasonal droughts in this region. Also, little is known about the distribution of fine and coarse tree roots growing below shallow soils into weathered bedrock, mainly because of difficulties in excavating bedrock.

This study responds to a cross-disciplinary call for the application and wide use of new geophysical methods to advance *in-situ* research in root ecology (Jayawickreme et al., 2014). Our study adopted a novel approach to tackle several questions simultaneously and by drawing upon diverse disciplines such as ecosystem ecology, ecohydrology, geophysics, and biogeosciences. Here, we present details on the application of surface geophysical imaging tools for root research studies in mixed forests in an edaphically, geologically and climatically extreme and complex environment. We emphasize the



application of these tools primarily for exploration and thus as an alternative and/or complement to traditional ecological methods to gather information on ecologically relevant subsurface variables across time and space. We expected weathered rhyolite bedrock in Sierra San Miguelito Volcanic Complex (SSMVC) to conserve humid microsites and that the root distribution of pine-oak forest stands mirrors spatial and temporal heterogeneity of water availability. In particular, we aimed at: (1) characterizing the presence and depth of weathered rock and demonstrating that exfoliation sites function as potential water sources; (2) with the use of GPR, detecting roots of different diameter size classes growing at various depths in volcanic fractured rock; (3) with the use of ERT, assessing the relationship between soil electrical resistivity and soil water potential in order to determine if resistivity tomography can detect the spatial and temporal variability of soil moisture beneath vegetation patches; and (4) with ERT tomograms, describing the functionality of weathered rock in forest ecosystems colonizing shallow rocky soils.

## 2 Materials and methods

### 2.1 Site description

The site is situated in SSM, in a semiarid pine-oak forest ecosystem in the Southern region of the SSMVC (Fig. S1). The SSMVC represents the remnants of one of the most voluminous rhyolitic volcanic events on Earth (McDowell and Keizer, 1977), formed by massive lava spills of rhyolitic composition (Portezuelo Latite and San Miguelito Rhyolite). Currently, this volcanic complex is affected by small-scale local fracturing through pedogenesis and hydrological processes and thereby is directly influencing pine-oak forest establishment (Fig. 1a-b). Soils are poorly developed and overall extremely shallow (<25 cm) and rocky; hence, to get support, tree roots commonly anchor in weathered rocks or beneath rock outcrops (Fig. 1c-d). Lithological profiles show a high density of vertical roots in rock fractures and soil pockets (Fig. 1e-f). According to the World Reference Base for Soil Resources (WRB) classification system, the soil at this site corresponds to lithic-paralithic Leptosols (LPlip) (FAO, 2006). Organic matter content is very high (60%) in these soils (Perez et al., 2009). The climate is semiarid; for the last 65 years mean annual precipitation (MAP) has averaged 408 mm (weather station "La Purisima", 22° 5' 22.4", 101° 12' 28.9"), where in 64% of the years MAP has been below average and only in 12% MAP has been above 500 mm (Fig. S2). In general, summer precipitation falls between July and October and accounts for 90% of MAP, the rest falls between December and February.

### 2.2 Experimental plots

Along a 2.5 km long transect running parallel to a narrow watershed, where pine and oak trees are evenly distributed in pure and mixed stands, we established a total of 12 circular experimental plots of 25 m diameter with four replicates per stand type (pine, oak, and mixed stands). In addition, for the exploration and tracking of roots with the GPR, we established one 8.5 x 6 m plot with parallel transects (spaced 1 meter) to observe horizontal axis elongation of roots in a mixed stand and one concentric plot of five circular transects around an anchored *Pinus cembroides* tree (with 0.1, 0.3, 0.5, 1.0, and 1.5 m





distance between neighboring transects). To monitor soil water potential ($\Psi_s$) at biweekly intervals, in September 2013, each circular plot was equipped with four (64 total) soil psychrometer sensors (TSP-55, Wescor Inc. USA), which were inserted at 12 cm depth (depending on the depth of soil pockets) near tree trunks (Table 2). To determine electrical resistivity, 72 geophysical electrodes (24 for each stand type) were installed with Northeast-Southwest orientation with 1 m inter-electrode

spacing (along the slope). GPR radargrams were generated using a MALÅ RAMAC ProEx GPR system coupled to an inspection wheel. Electric Resistivity Tomography (ERT) tomograms were taken using the SYSCAL KID SWITCH-24 (IRIS instruments) with a 24-multi-electrode switch box.

### 2.3 Principles of GPR

GPR is an impulse radar system designed for shallow subsurface investigations at 1 to 25 m depth. A transmitting antenna of

a certain frequency sends electromagnetic pulses from the soil surface through the soil matrix; a boundary layer is reflected when the transmitted pulse crosses two objects of different electromagnetic properties (Van Dam, 2014). Consequently, the reflected wave returns to the receiving antenna at ground level (or soil surface), which measures the reflected signal as a function of time (Butnor et al., 2001). Reflections and diffractions of electromagnetic waves may occur at boundaries between rock strata and objects that exhibit differences in electrical properties. Most soils and rocks have extremely low

conductivity (about $< 10^{-2}$ S/m), thus the propagation of electromagnetic waves is mainly affected by electrical dielectric constants of soils and rocks (Heggy et al., 2003). Electric permittivity, $\mathcal{E}$, and electric conductivity, $\sigma$, are petrophysical parameters that determine the reflectivity of boundary layers and penetration depth. Generally, the reflection of an electromagnetic wave occurs at boundary layers and its strength is shown by the reflection coefficient, $r$, which is determined by (Blindow et al., 2007):

$$r = \frac{\sqrt{\varepsilon_1} - \sqrt{\varepsilon_2}}{\sqrt{\varepsilon_1} + \sqrt{\varepsilon_2}}$$

In this equation, $\mathcal{E}_1$ and $\mathcal{E}_2$ are the dielectric constants of roots and soil, respectively. Specifically, the contrast in dielectric constants between a root and the surrounding soil determines root radar reflectance (Fig. S3a). The larger $r$ and the stronger the reflected wave at the boundary layer, the size and bow of the resulting hyperbola vary according to the amplitude of the reflected wave. The difference in dielectric permittivity of a root and its surrounding matrix forms a

boundary, which then can be detected by a traveling electromagnetic wave; however, it varies in time and space as a function of soil (texture, water content) and root characteristics (size, depth, orientation, water content). When a traveling electromagnetic wave hits a boundary between materials with differing electromagnetic properties, such as dry soil and water-conducting roots, part of this wave is reflected (Raz-Yaseef et al., 2013) often producing hyperbolic patterns (Butnor et al., 2001). We have worked with waveform parameters of the time interval between zero crossings (ns) of maximum and

minimum reflected waves (Guo et al., 2013).





## 2.4 Principles of ERT

ERT is a method that produces images of the variation of electrical resistivity in either two or three dimensions, below a line or grid of electrodes placed on the soil surface. ERT tomograms consist of a modeled cross-sectional plot of resistivity ($\Omega \cdot m^{-1}$) versus depth. The method is based on measurements of voltage differences between electrodes. This is a minimally

invasive method, because it only requires inserting electrodes a few centimeters into the ground to create electrical contact. The resulting subsurface resistivity model depicts variations in the conductivity of electrical current in subsurface soils and rocks (Fig. S3b). Resistivity is the mathematical inverse of conductivity. The measured resistivity is a function of water content of the substrate (rock or soil), the chemical composition of pore water and the soil surface area/grain particle size distribution. The relations of these variables are summarized in Archie's law, an empirical equation of resistivity, $\rho$ [$\Omega \cdot m^{-1}$],

of rocks (König et al., 2007):

$$\rho = \frac{a}{\phi^m S^n} \rho_w$$

where $\Phi$ (porosity) and $S$ (saturation factor) are fractions between 0 and 1, $\rho_w$ [$\Omega \cdot m^{-1}$] is the resistivity of groundwater, and the parameters $a$ (tortuosity), $m$ (cementation factor), and $n$ (saturation exponent) are empirical constants that need to be determined for each study area.

## 2.5 Data collection

During October to December 2012, we examined the frequency, size, position and depth range of roots with 2.5 to 7.5 cm diameter in organic soil and under exfoliated rocks using the GPR 500 MHz antenna. To characterize the exfoliation of weathered rocky soil, and to differentiate between the exfoliated rock base and potential root axes (0.6 to 4 cm diameter) underneath rocks, we used the GPR 800 MHz antenna. Root identification in radargrams was a stepwise process; first, roots

were recognized at locations where hyperbolas of reflected waves had relatively higher amplitudes compared to those in the surrounding area (Cui et al., 2011). Then, to determine the diameter and depth of these roots, the time interval between zero crossings (ns, time interval for maximum reflected wave) was extracted at the points of hyperbolas, where roots had been identified previously. The detection frequency for the number of roots identified in the radar profile was calculated along each transect for five root diameter classes (< 3.0, 3.0 – 4.0, 4.0 – 5.0, 5.0 – 6.0, and > 6.0 cm). Finally, for calibration

purposes, individual roots (total of 76) were excavated to determine their depth and diameter *in-situ* (Table 3, Fig. 2a).

Based on the assumption that electrical resistivity decreases with increasing water content (Nijland et al., 2010;Jayawickreme et al., 2014), we compared the spatial and temporal patterns of soil and rock moisture within and among forest stands. During an 8-months period, we generated 12 ERT tomograms (four for each stand type) during wet (October 2013 and February 2014) and dry (December 2013 and May 2014) ecohydrological periods. Here we present one

representative profile for each forest stand. To relate $\Psi_s$ data with ERT surveys, $\Psi_s$ was measured at diurnal peaks of water stress (from 11 to 14 hrs) every two weeks during an 8-months period (October 2013 to May 2014).





Finally, to trace short-term percolation responses and the advancement of water profiles (March 2013) during the dry season, 15 liters of water were injected into a shallow fracture of 35 cm length in an oak stand; 150 minutes later, we generated radar profiles along a 3.3 m transect running parallel to the slope with the 800 MHz antenna.

**2.6 Data processing**

Raw GPR radargrams were processed with RadExplorer v1.42 software (Mala GeoScience, USA Inc) prior to interpretation. Filtering of radar data removed unwanted signals (noise) and corrected the position of reflectors on the radar record. The sequence of filter application depends on the accuracy of collected radargrams and the study's objective. For each particular case, radargram processing follows specific procedures (Guo et al., 2013). In this study, for root exploration all radargrams were processed with the same range of filter values, because the whole study area had similar characteristics with a horizon

of organic soil and weathered rock underneath. The background removal filter eliminates parallel bands resulting from plane reflectors such as ground surface, leaf litter, soil horizons (when it comes to identifying roots in the soil), and bands of low-frequency noise (Butnor et al., 2003;Dahboosh Al-Shiejiri, 2013). Stolt F-K migration routine was used to correct for object position and collapsed hyperbolic reflections (diffracted waves) based on signal geometry (Dahboosh Al-Shiejiri, 2013). The waveform parameter of the time interval between zero crossings (ns) of the maximum reflected wave was extracted at the

points of root detection in the radar profiles and calculated using RadExplorer v1.42 software.

Electrical resistivity tomography was conducted using a wenner-switch array. Resistivity values were corrected for the effect of temperature, based on the temperature recorded by the closest soil psychrometer sensor at a given depth for each resistivity value, and on the Campbell equation (Campbell et al., 1949) as suggested by Samouëlian et al. (2005):

$$\rho = \rho T[1 + \alpha(T - 25)]$$

where,

$T$ corresponds to temperature (∘C), $\rho T$ is the electrical resistivity measured at temperature T ($\Omega \cdot m^{-1}$), $\rho$ is the electrical resistivity at the reference temperature of 25 ∘C ($\Omega \cdot m^{-1}$), and $\alpha$ refers to a correction factor equal to 0.0202.

Inversion and forward simulations were performed with RES2DINV software (Geotomo software) for later manipulation of data files with the ArcMap module applying an Empirical Bayesian Kriging method (ArcGIS Desktop,

ESRI 2011). For more details on the softwares and algorithms used see Krivoruchko, 2012 and Loke, 2015.

**2.7 Statistical analysis**

Nested two-way analysis of variance was used to examine differences in root diameter. The model included two factors, forest stand with three levels (pure and mixed pine and oak stands; fixed effect) and soil depth with four levels (10, 20, 30

and >30 cm; nested effect); in case of significant interactions we conducted Tukey's *post hoc* mean comparison test. We ran Type I regression analyses to examine the relationships between root diameter and time interval between zero crossings (ns) for both frequency antennas to calibrate the method. Polynomial quadratic regression analyses were conducted to examine



the relationship between $\mathit{\Psi}_{soil}$ (MPa) and resistivity (Ω·m$^{-1}$). Prior to statistical analysis we applied Shapiro Wilk's test to examine normality of the residuals. All statistical analyses were run in SAS University Edition (Free Statistical Software).

## 3 Results

All geophysical images helped interpret the spatial distribution of tree roots in soils and rocks, as well as of rocky soil characteristics. However, some difficulties in the interpretation of raw radargrams (unfiltered radar profiles) included noise and ghost areas caused by characteristics of organic and rocky soils. Nevertheless, radargrams indicated clear hyperbolic reflectances that corresponded to the position of tree roots at certain depths (Fig. S3a). ETR tomogram results (Fig. S3b) for the top 50 cm helped identify areas of greatest drainage (200-450Ω·m$^{-1}$) and fracturation (400-700Ω·m$^{-1}$). ETR tomogram outputs of RES2DINV software did not reveal the exfoliated rocks that occur in the study area.

### 3.1 GPR detection of tree roots and diameter estimation

We examined the relationships between root diameter and time interval between zero crossings (ns) using 500 MHz signals ($P < 0.0001$, $R^2 = 0.93$, Table 4, Fig. 2a). With this antenna, we detected roots as fine as 2.5 cm diameter and as coarse as 7.5 cm in different tree stands. In pure pine stands, the finest roots (2.5-3 cm) were preferentially located in the top 10 cm of the organic soil, while roots with coarse diameters occurred mostly at 30 cm depth (Table 5, Fig. 2b). In contrast, in pure oak stands root diameter decreased with increasing soil depth (Fig. 2b). Also, in mixed stands, deeper roots had overall smaller diameters (Fig. 2b) than roots in shallow soils. It is important to remark that in pure pine stands no roots occurred below 30 cm depth. Fig. S4, depicts a typical radargram generated with a 500 MHz antenna after having applied band pass filtering and the background removal filter. The GPR radargram of a mixed forest stand reveals highest aggregation of coarse roots near the tree bases and their adjacent areas, as well as a high heterogeneity of root diameters distribution between trees (Fig. S4). Radargrams also revealed a clear boundary layer between soil and rocky substrate (Fig. S4: continuous line) and soil pockets (Fig. S4, dotted line). These soil profiles were validated *in situ*.

In a mixed stand, a two-dimensional radargram sequence was generated with the 500 MHz antenna in an 8.5m x 6m horizontal tracking quadrant; both pine and oak roots (different uppercase letters in Fig. 3) were validated *in-situ*. For this serial root mapping consisting of 7 sequential radargrams, we identified a total of 386 roots in their horizontal position. Diameters of single roots were highly variable (2.5-6 cm), as well as signal outputs (hyperbolas) for deep roots (5-30 cm). With this sequential series of radargrams, we could track the horizontal placement (elongation) of single root axes (for instance root "B" in the different radargrams) (Fig. 3b - g). Surprisingly, we found high variation in root diameters along a 6 m transect (root diameters in radar profile a = 6.5, b = 4.8, c = 5.2, d = 5.8, e = 6.4, f = 4.6, g = 3.8 cm) in accordance with the size of signal amplitude (in radar profile a = 1.16, b = 0.78, c = 0.84, d = 0.94, e = 1.09, f = 0.73, g = 0.61 ns). Roots with larger diameters had higher signal amplitudes, whereas smaller diameter roots had lower amplitudes.

We identified pine trees anchored under exfoliated rocks by applying 800 MHz antenna along concentric transects around a tree and after adopting background removal routines (Fig. 4). We observed a significant relationship between root



diameter and time interval between zero crossings using 800 MHz signals ($P < 0.0001$, $R^2 = 0.97$, Table 4, Fig. 2a). The radargram also revealed (Fig. 4b, uppercase letters) roots under exfoliated rocks, which were then used to calibrate the radargram. This technique allowed us to differentiate between the base of exfoliated rocks (about 35 cm deep) and the presence of thin roots underneath that rock (0.6 to 4 cm in diameter). In the transect at 50 cm distance from the trunk, three

hyperbolas were reflected, suggesting root presence under the exfoliated rock (Fig. 4b). By increasing transect length, the number of reflected hyperbolas increased under the rock (e.g., transect at 100 cm distance from tree base) and the rock limits, permitting to track root elongation in shallow rocky soils. Also, it was possible to match a high signal amplitude with a pine tree anchored beneath an exfoliated rock (Fig. 4b, letter "B") and its associated lateral roots (Fig. 4b, arrows). In spite of filtering routines, it is impossible to completely remove all noise sources in all radargrams; e.g., there was some residual

noise associated with leaf litter accumulation under exfoliated rocks, however *in situ* verification confirmed that radargrams spotted primarily tree roots.

### 3.2 Distribution of roots and subsurface resistivity imaging

ERT tomograms of different forest stands revealed a clear horizontally layered structural organization of weathered rock with exfoliation (Fig. 5). The tomograms of 0-2 m depth showed a wide range of resistivity values with maxima >1000 $\Omega \cdot m^{-1}$

observed at tomogram bottoms and a minimum between 250 and 650 $\Omega \cdot m^{-1}$ at surface horizons. In the upper horizons (< 0.5 m), the observed low resistivities (< 450 $\Omega \cdot m^{-1}$) corresponded to islands of higher root densities beneath vegetation patches and were associated with water extraction zones; high resistivities (> 450 $\Omega \cdot m^{-1}$) coincided with bedrock outcroppings. Considering vertical distributions of pine and oak roots (Fig. S4), ERT tomograms clearly matched root distribution to species-specific vegetation cover at the measurement points.

Following seasonal drying (Dec 2013 to May 2014), ERT profiles of all stands exhibited increasing resistivity at 0 to 1 m depth (Fig. S5), which was likely attributed to soil drying as a consequence of both root water uptake and soil evaporation. Thus, in the top meter, we observed the largest spatial variation in rock moisture content ranging from 250 - 1450 $\Omega \cdot m^{-1}$ ($\Psi s$ = -0.5 to -24.5 MPa). By visual assessment, both mixed and pure oak stands showed highest moisture content in all four monitored periods (Fig. S5e - l). Pure pine stands preferentially occurred on sites with deepest soils (up to

60 cm), while pure oak stands anchored mostly on exfoliated rocks. Mixed stands had a combination of both abiotic site characteristics.

Soil resistivity and soil water potential were negatively related considering all observations from the different stands ($R^2 = 0.95$; $P < 0.0001$, Fig. S6). Thus, as resistivity increased $\Psi s$ dropped; this trend was apparent for resistivity values up to around 750 $\Omega \cdot m^{-1}$ being proportional to almost -6 MPa.

### 3.3 Fracturing as a secondary water supply to forest trees

In the dry season, upon water injection into a rock fracture close by an oak tree base growing on exfoliated rock (Fig. 6a), we observed a response signal in form of a wave amplitude equivalent to those observed underneath vegetation patches (Fig.



6b). Radargrams showed a clear infiltration horizon at approximately 50 cm depth. The signal appeared 150 minutes after water injection; while it was not homogeneous for all vegetation patches, because of different root densities and other microsite differences, we could detect a remarkably rapid horizontal displacement of water up to three meters distance from where it was injected.

## 4 Discussion

In semiarid environments, forest ecosystems that develop on young volcanic bedrock and poorly developed soils face two independent growth limitations, 1) highly variable precipitation and increasing frequency of droughts, and 2) extremely low water storage capacity of soils. Hence, insight into the distribution of different tree root types at the soil-rock interface and the spatio-temporal availability of water is fundamental for understanding tree ecophysiology, tree population dynamics, tree species interactions, and forest ecosystem functioning. Lack of instruments and technology to study belowground root ecology has delayed scientific advances in forest ecosystem ecology in semiarid regions. However, with interdisciplinary efforts and the employment of geophysical tools and standard methods in ecosystem science (e.g., use of natural abundance and tracer stable isotopes) potentially great advances may be achieved in our understanding of the underlying geoecohydrological mechanisms that may explain tree species coexistence in extreme water-limiting environments (Rodriguez-Robles et al., 2015). Here, we demonstrate the enormous potential of applying geophysical tools to examine non-destructively and in real-time soil-rock-water and root characteristics.

With the use of GPR, we clearly detected pine and oak roots with diameters ranging between 0.6 and 7.5 cm under natural soil conditions (root diameters of 2.5 to 7 cm and 0.6 – 4 cm with the 500 MHz and 800 MHz antenna, respectively; Figs. 2 and Appendix S1: S4). Typically belowground studies using GPR are carried out in sites with homogeneous soils, such as forest plantations, gardens, parks, backyards, crop fields, or under highly controlled conditions (Ow and Sim, 2012;Cermak et al., 2000;Cox et al., 2005;Dannoura et al., 2008;Zenone et al., 2008;Zhu et al., 2014) to reduce the difficulty in detecting and interpreting the origin of reflected signals (hyperbolic). In our case, it was fundamental to use high and low frequency antennas, as they gave valuable complementary information on these complex shallow rocky soils over volcanic rocks. With the 500 MHz antenna, we could differentiate between and characterize a series of vertical substrate layers, whereas with the 800 MHz antenna we could locate thin roots underneath exfoliated rocks. However, with the 800 MHz antenna, detection efficiency of fine roots decreased in sites with high litter accumulation of fresh pine needles on exfoliated rocks (Fig. 4). Similar difficulties for GPR interpretation had been mentioned previously; for instance (Hirano et al., 2009) reported that soil water content may greatly limit the detection of reflected waves originating from roots. In October 2013 (a wet month), we carried out an experiment with the GPR 500 MHz to examine the wetness effect in one of the experimental plots. Under high soil moisture content, we did not get a signal from the roots at this site, most likely because the signal wavelength gets lost by undetected changes in the dielectric properties between roots and soil (material interface) (Guo et al., 2013;Hirano et al., 2012;Butnor et al., 2009). Although our GPR survey was carried out in the dry period, we were still facing some difficulties to accurately differentiate between hyperboles deriving from different yet overlapping roots. The





experiment of local water injection in a rock fracture (Fig. 6), greatly helped to clearly identify and separate the hyperbolas (roots) in the radargram.

## 4.1 Dynamics of water inside of weathered rocks and spatial distribution of roots

In forests colonizing shallow soils and impermeable volcanic rock, water availability largely depends on the soil-weathered rock interaction (Fig. 5). In one particular case, water of an accumulated 87 mm rain event occurring in February, infiltrated and percolated down to only 50 cm depth (Fig. S5 c, k). Although volcanic rock is characterized by low permeability, rock fracturation may contribute to what can be called "secondary substrate porosity" in impermeable rock, thereby allowing water-flow and storage within volcanic rock (Carrillo-Rivera et al., 1996). The analysis of ERT tomograms revealed a clear detachment of rock layers (exfoliation) (Fig. 5) and the presence of soil pockets (Fig. 5b, at location 18.4 m of the transect), which are both formations that potentially favor water storage. These conditions appear to promote the establishment and anchorage of trees under otherwise highly limiting soil water conditions. Several studies have reported that trees can get established in rock fractures (mainly calcic and permeable rocks) (Estrada-Medina et al., 2010;Poot and Lambers, 2008) and that they locate their roots inside of this permeable material to exploit stored water (Querejeta et al., 2007;Schwinning, 2013). The combination of tomogram and radargram images (Fig. 5) reveals distinct microsites in this shallow layer of weathered rock suitable for tree species establishment and the formation of vegetation patches. Also, our interpretation of images suggest that oak and pine might exhibit complementary strategies to access different water sources. Oak distributes its finest roots in both the soil organic layer and in the soil-weathered rock interface (Figs. 2b, 5b). This rooting pattern may enable oak to access water retained in weathered rock during the dry periods (Fig. 6). Pine, in contrast, absorbs water exclusively from shallow surface soil (Figs. 2b, 5a). Species-specific differences and preferential horizontal (pine) and vertical (oak) root distribution in these geohydrological niches suggest the two species coexist in these ecosystems (Rodriguez-Robles et al., 2015). Additional studies are needed to attest this possibility.

Our assay of injecting water into a rock fracture in the dry period showed that oak roots responded rapidly, i.e. within 150 minutes, to a short-term water pulse, which moved 300 cm laterally, suggesting some sort of channel type connection between fractures and exfoliated rocks (Fig. 6). Hence, exfoliated rocks may play key bi-functional ecological roles: they support tree anchorage (Fig. 4a) and serve as vital water entry, reservoir and distribution points during dry periods (Fig. 6b). Root anchorage in exfoliated rocks at this site can be considered as a survival strategy, since trunks and horizontal roots located below exfoliated rocks obtain physical support, which cannot be provided otherwise in these particularly shallow soils (Fig.5).

## 5 Conclusion

This study highlights thus far underexplored yet potentially extremely powerful tools of geophysical imaging in forest ecohydrology. They allow *in situ* non-destructive estimation of a wide range of tree root diameters, with 0.6 cm as the highest resolution of diameter and the location of short-term and long-term water reservoirs in a complex soil – rock terrain.




Furthermore, non-invasive mapping of GPR and ERT provides detailed field-level information of geohydrological characteristics of the soil - weathered rock interface, which were traditionally assessed with coring and excavation methods. This study demonstrates that the application of ERT and GPR has an enormous potential to capture belowground spatial and temporal characteristics of roots and soil moisture distribution at the field scale.

While these tools offer many advantages for the study of belowground *in situ* aspects of ecosystems and RadExplorer and ArcGis software allow powerful image processing and manipulation of radargrams and tomograms, we want to highlight the major limitations we encountered in this study; certain field conditions (e.g., leaf litter, regolith) i) impede or reduce the detection potential and quantification of coarse roots when using the GPR 500 MHz antenna; ii) they also reduced the capability of the GPR 800 MHz antenna to delineate reflection signals emitted by roots; iii) an increase in

soil moisture may decrease the electromagnetic gradient between roots and soil, such that reflected signals get considerably weakened, which makes root delineation more difficult under wet conditions; iv) given the shallow rocky soil contact resistance problems may occur, especially during dry periods. We minimized these problems by pre-cleaning the surface of litter and twigs (points i and ii) and by applying copper sulfate gel in the inserted electrodes (point iii).

     Geophysical images are highly valuable and promising tools to advance our understanding of the coupled nature of

15 geoecohydrological patterns and processes by linking belowground geophysical structures with soil/rock hydrological characteristics and root ecology.

## 6 Author contribution

U.R.-R., J.T.A. and J.A.R.-L. planned and designed the research and executed the field experiments. U.R.-R., J.T.A., E.H.-S. and E.A.Y. analyzed the data and wrote the manuscript.

## 20 7 Acknowledgements

We thank Alejandro Cruz Rosas Palafox for technical advice in the use of the ERT equipment. We thank the Applied Geosciences Division at IPICYT for providing access to geophysical instruments and tools. URR acknowledges the National Council for Science and Technology of Mexico (CONACyT) for the scholarship no. 332356 granted to pursue his PhD degree. This work was supported by grants given to JTA from CONACYT, num. 220788 and 224368.

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



**Table 1.** A cross-study comparison of the detection capacity (minimum and maximum) of tree root diameter and depth in different soil types using GPR systems with various radar frequencies.

| Radar frequency (MHz) | Tree species | Soil type | Site Condition | Detected root diameter (cm) | | Detected root depth (cm) | | Reference |
|---|---|---|---|---|---|---|---|---|
| | | | | min | max | min | max | |
| 400 | *Pinus taeda* | Gergeville soil | Plantation | 3.7 | 10 | - | 130 | *Butnor et al., 2001.* |
| 400 | *Melaleuca quinquenervia* | Flat sandy soil | Controlled, root segments | 3 | 13 | - | 50 | *Nga et al., 2014* |
| 450 | *Quercus petraea* | Loamy deluvial soil | Plantation | 3 | 5 | - | 200 | *Hruska et al., 1999.* |
| 500 | *Larix kaempferi* | Forest soil | Plantation | - | 10 | 10 | 30 | *Zhu et al., 2014.* |
| **500** | ***Quercus potosina, Pinus cembroides*** | **Shallow, rocky soils** | **Semiarid tropical forest** | **2.5** | **7.5** | **3** | **40** | **This study** |
| 500 | *Ulmus pumila, Artemisia ordosica* | Dry sandy | Controlled, fixed sand dunes | - | 3.5 | 10 | 80 | *Cui et al., 2011.* |
| 800 | *Eucalyptus sp.* | River sand | Plantation | - | 5 | - | 50 | *Barton & Montagu, 2004(Barton and Montagu, 2004).* |
| **800** | ***Quercus potosina, Pinus cembroides*** | **Shallow, rocky soils** | **Semiarid tropical forest** | **0.6** | **4** | **1** | **40** | **This study** |
| 900 | *Prunus persica* | Faceville fine sandy loam | Controlled, peach orchard | 2.5 | 8.2 | 11 | 114 | *Cox et al., 2005.* |
| 900 | Different tree species | Red-yellow and marshy soils | Subtropical evergreen forest | 1 | 3 | 1 | 60 | *Yan et al., 2013.* |
| 1,000 | *Eucalyptus sp.* | River sand | Plantation | 1 | 10 | - | 50 | *Barton & Montagu, 2004* |
| 1,000 | *Quercus douglasii, Pinus sabiniana* | Auburn-exchequer soil | Semi-arid savanna | 1.3 | 10 | 8 | 30 | *Raz-Yaseef et al., 2013.* |
| 1,500 | *Populus deltoides* | Lakeland soil | Plantation | 0.6 | 1.7 | 11 | 27 | *Butnor et al., 2001.* |
| 2,000 | *Ulmus pumila, Artemisia ordosica* | Dry sandy | Controlled, fixed sand dunes | 0.5 | 3.5 | - | 30 | *Cui et al., 2011.* |




**Table 2.** Number of trees per stand and species included in the study of soil water potential (*n*), as well as mean or tree diameter at breast height (DBH) and total tree height of trees of *Pinus cembroides* and *Quercus potosina* at Sierra San Miguelito, San Luis Potosí, Mexico.

| Stand | *n* | DBH (cm) | Tree height (m) |
|---|---|---|---|
| Pine/pure | 16 | 18.701 ±2.49 | 4.863 ±0.74 |
| Oak/pure | 16 | 21.104 ±1.67 | 5.272 ±0.86 |
| Pine/mixed | 16 | 19.981 ±1.76 | 6.080 ±1.17 |
| Oak/mixed | 16 | 20.121 ±1.38 | 5.461 ±1.08 |

**Table 3.** Calibration and validation of roots (diameter and depth). This table represents the information extracted from a radargram of a pine-oak stand, Fig. S4.

| Along transect | *In situ* | | GPR 500 MHz | |
|---|---|---|---|---|
| (m) | diameter (cm) | depth (cm) | diameter (cm) | depth (cm) |
| 1.35 | 2.6 | 7.4 | 2.8 | 7.2 |
| 1.68 | 2.7 | 9.4 | 2.8 | 9.6 |
| 3.18 | 2.5 | 15 | 2.4 | 15.2 |
| 3.76 | 2.6 | 22.4 | 2.8 | 23.1 |
| 3.98 | 3.7 | 8.5 | 3.6 | 8.9 |
| 4.85 | 6.7 | 13.5 | 7.0 | 13.9 |
| 5.35 | 2.9 | 21.5 | 3.2 | 22.2 |
| 6.90 | 4.4 | 9.4 | 4.8 | 9.9 |
| 7.12 | 2.8 | 13.8 | 2.6 | 14.2 |
| 8.56 | 4.7 | 12.4 | 5.0 | 13.0 |
| 10.78 | 2.5 | 13.0 | 2.8 | 13.4 |
| 11.92 | 3.4 | 12.2 | 3.0 | 11.8 |





**Table 4**. Intercepts, slopes, regression coefficients and observed probabilities of linear regressions between root diameter (cm) and time interval (ns) which were used for calibration with both GPR systems.

| GPR systems | Intercept ± 1SE | Slope ± 1SE | $R^2$ | $P$ |
|---|---|---|---|---|
| 500 MHz | 0.1964 ±0.2006 | 5.7482 ±0.2223 | 0.94 | <0.0001 |
| 800 MHz | -2.0910 ±0.1273 | 6.2450 ±0.1839 | 0.98 | <0.0001 |

**Table 5.** Nested two-way analysis of variance for examine differences in root diameter for four soil depth (10, 20, 30 and >30 cm) comparing forest stand (*Pinus cembroides*, *Quercus potosina* and Mixed) in a semiarid forest ecosystem in Central-North México.

| Effect | df | $F$ | $P$ |
|---|---|---|---|
| Stand | 2 | 8.51 | 0.0002 |
| Depth (Stand) | 8 | 184.98 | <0.0001 |

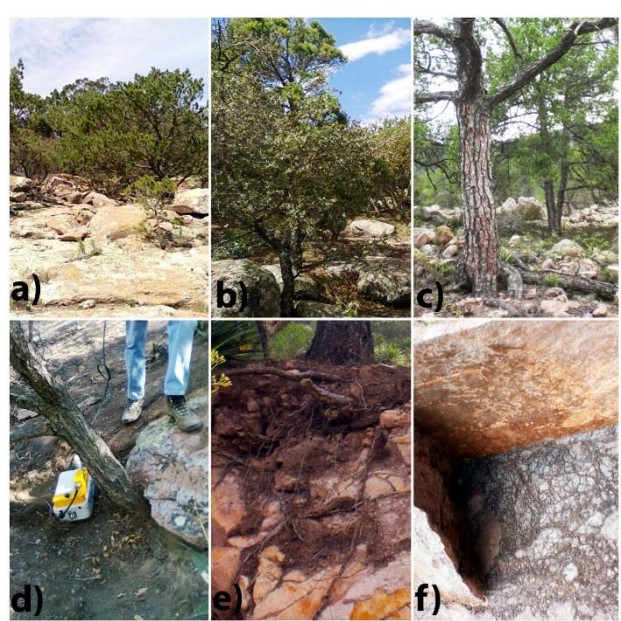

**Figure 1**. Site characteristics: (a) rhyolitic weathered rock in Pinus cembroides stands, (b) exfoliated rock in a pine-oak stand, (c) exposed coarse roots of 14 cm diameter in shallow rocky soils, (d) pine anchored under exfoliated rock, (e) high

10  root density at the soil/bedrock interface, (f) fine roots colonizing a weathered bedrock layer.




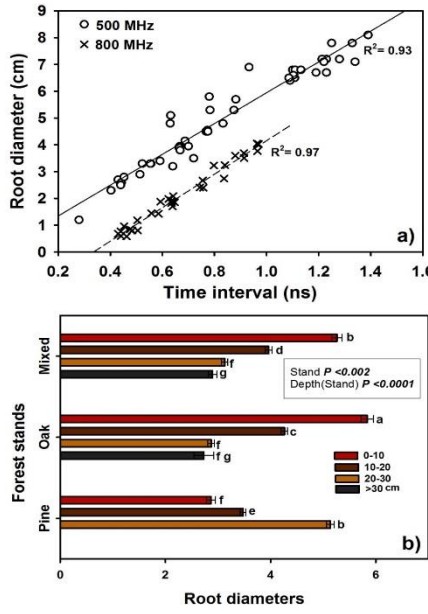

**Figure 2.** (a) Relationship between root diameter from different stands and time interval with zero crossing of detected roots, which were later used for calibration with both GPR systems: 500 MHz frequency antenna ($n = 48$), $P < 0.0001$, and 800 MHz frequency antenna ($n = 28$), $P < 0.0001$. (b) Average diameter of roots recorded by GPR for each forest stand type at four depths. Different letters next to bars indicate statistical differences at a probability value of $P < 0.005$.

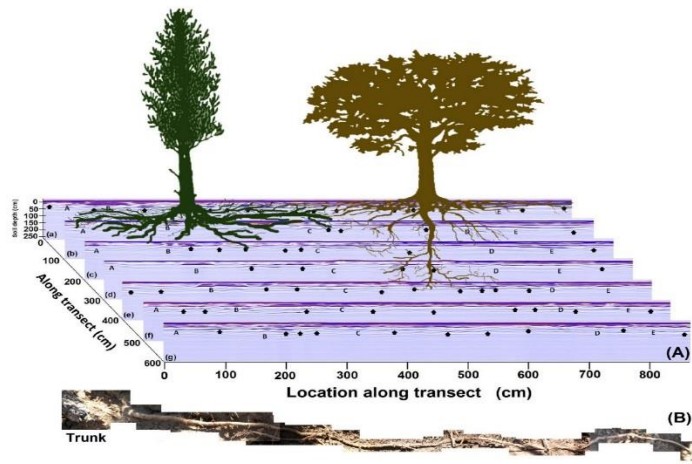

**Figure 3.** Mapping root systems with GPR show the potential for radargrams to represent an approximation of horizontal root distribution: (A) 2-D radargram sequence obtained with the 500 MHz antenna performing seven parallel transects of 8.5




meters, spaced 1 meter apart. In each radargram, cross-sections of roots were identified and then their diameters estimated. By linking root reflectors in neighboring GPR radargrams, the orientation and length of each single root were obtained (same letters). Arrows and letters in different GPR radargrams correspond to reflections from the same roots; they were used for calibration in situ. With the 500 MHz antenna the position, size, and depth of roots with 2.5 to 7.5 cm diameter were

5    estimated. The image shows an example of the position of a pine and oak tree and the potential application of the GPR tool for spatially explicit root distribution studies. (B) Horizontal elongation of a root axis of Pinus cembroides marked with the letter "B" in (A).

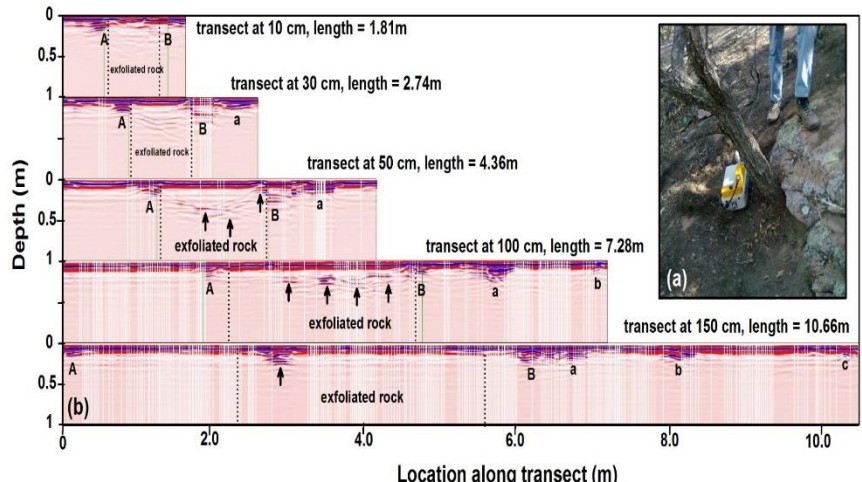

**Figure 4.** Concentric transects used to detect and track lateral root proliferation with GPR: (a) In situ photography showing a Pinus cembroides root anchored in exfoliated rock and the GPR system with the 800 MHz antenna. Five circular transects were established around the tree with 0.1, 0.3, 0.5, 1.0 and 1.5 m distance between neighboring transects. The radius of each transect varied from 0.29 to 1.7 m; (b) Corresponding GPR radargrams of different transect lengths. Same letters in different

15   GPR radargrams indicate examples of reflections from the same roots; uppercase letters indicate roots used for calibration. Arrows indicate root presence under exfoliated rocks.





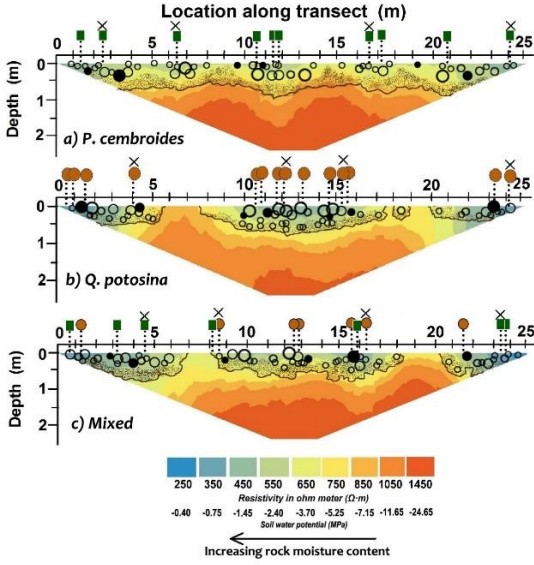

**Figure 5.** ETR tomograms in (a) Pinus cembroides, (b) Quercus potosina and (c) pine-oak forest stands. ERT profiles showed a relationship between the position of roots, low soil resistivity (greater water availability) and greater rock fracturing. The top layer corresponds to the soil layer, the intermediate layer to pockets of soil and rock fractures and the bottom layer represents fresh rock. Circles of different size depict roots of different diameter size (see figure legend in Fig. S4). Black circles indicate roots that were used for GPR calibration. Trees marked with X indicate the presence of soil psychrometers sensors.

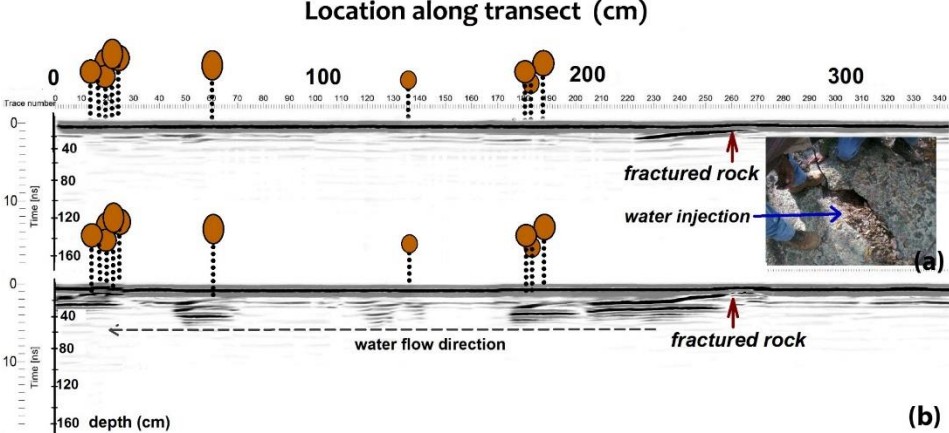

**Figure 6.** GPR radargrams showing how oak roots are preferentially located in fractured rocks where the probability of water accumulation is high. (a) GPR radargram in dry condition. (b) GPR radargram 150 minutes after the injection of 15 L of water in a rock fracture. In the radargrams filters were applied to highlight areas of interest. Inserted photo: rock fracture, where water was injected