# Peer review of "Application of geophysical tools for tree root studies in forest ecosystems in complex soils"

_Biogeosciences, 2017_

## Referee Comment (RC1) · V. Resco de Dios (Referee) · 2 May 2017

Rodríguez-Robles et al present the results of jointly using GPR and ERT for study soil water and root dynamics. I cannot evaluate this as an expert in these geoscientific tools, but as a scientist with a general interest in water uptake and a potential end-user in the long term. Considering this perspective, I found the manuscript provides a set of novel tools to the vexed problem of assessing root water uptake, particularly in semi-arid, rocky environments. The authors show how this is the first study using this technique shallow and rocky soils, rendering the manuscript as novel. The graphics and overall data presentation is outstanding (although axes labels could be made bigger for ease of reading). I only have a couple of minor comments, in case they help to

improve even more this study: 2.2 – PST 55 (notice the typo) are notorious for lack of temperature compensation. How did you account for that? Table 5 – please pay attention to wording of legend. Please clarify x-axis and legend in fig. 2b. Note also that roots are thicker at the top... does the results of thicker roots in pine at 30cm indicates that 1 single big root at the top divides into more thick roots afterwards? How do you explain having more thick roots at deeper surfaces for pines? That's particularly surprising because , as your diagram in Fig. 3 shows, pines have a shallow root system Fig. 5 – there are some fairly low soil water potentials. Why?

---

## Referee Comment (RC2) · Anonymous Referee #2 · 8 May 2017

In my opinion, this is an interesting, original and apparently technically sound study. However, please note that I am not an expert on the technicalities and hands-on details of GPR and ERT techniques and data analyses, so I strongly recommend additional review of the more methodological aspects of the study by a real expert on these geophysical techniques. The paper is concise and very well written, and I think that it represents a valuable contribution that significantly advances current scientific knowledge in a clearly underrepresented field in the literature (i.e. vegetation rooting and water uptake patterns in ecosystems with very shallow soils). At the same time, I appreciate that the Discussion section ends with an honest acknowledgement of the caveats, limitations and technical problems encountered with the application of GPR

and ERT techniques in the field.

L23 in Abstract: There is a typo in the spelling of ERT.

L3, page 5: please be more precise (you mean that sensors were inserted at 12 cm depth only when allowed by bedrock?)

L18, page 6 and elsewhere: Please consider the option of using the terms "bedrock" and "weathered bedrock" (instead of just "rock") whenever appropriate, and please replace throughout the text.

Line 10, page 9: leaf litter accumulates under exfoliated rocks, or on exfoliated rocks (L26, Page 10), or both?

Lines 14-17 in page 8 and again in Lines 16-18 in page 11: this appears like an unusual rooting pattern, did you find any other published papers reporting similar rooting patterns in oaks or in other tree species? Can you provide any references of similar findings?

Line 27-28 in page 8: Why is this finding so surprising, am I missing something here...?

Line 10, page 9: only leaf litter accumulated under rocks? what about root litter, was it also present under rocks? Were you able to distinguish between living and dead roots?

Line 7, page 12: Please clarify what exactly is meant by regolith here (wheathered bedrock only? all types of bedrock?).

For the sake of clarity, please rephrase item IV in lines 11-12 of page 12, this sentence is a bit confusing.

The quality of the figures is rather good, although I have some suggestions for improving Fig 5 (whose size in the final version should be at least twice as big as that in the PDF version that I have reviewed). I was a little confused by the legend of Fig 5, as it is not clear to me whether the three layers mentioned in the legend (soil, intermediate, bottom) are depicted or delineated in any way in the figure or not...It appears that

[Figure]

only the water potential categories are represented by different colors. Also, please note that some of the "soil" water potential values shown in this figure (-24 MPa) are extremely and unusually low for soil and require further clarification. I guess these values represent water potential values for dry bedrock, not soil. Also, applying the terms "Increasing rock moisture content" and "soil water potential" to the same moisture potential data appears rather contradictory. For the sake of clarity, I recommend to change to "Soil/bedrock water potential" and "Increasing soil/bedrock moisture content".

Figure 3 should also be enlarged in the final version of the published paper. With the current size, it is very difficult to spot the B (in A) mentioned in the legend...

Table 5: four soil DEPTHS....comparing forest STANDS...

The correct reference for Querejeta et al (2007) is:

Querejeta JI, Estrada-Medina H, Allen MF, Jimenez-Osornio JJ (2007) Water source partitioning among trees growing on shallow karst soils in a seasonally dry tropical climate. Oecologia 152:26–36

―――――――――――――――――――

---

## Author Comment (AC1) · 29 May 2017

Dr. V. Resco de Dios Referee (RC1)

We have now reviewed the comments by RC1. We appreciate the suggestions and comments posed by the referee. Below we include the answers to all his questions in this version of the manuscript.

Comment 1. Rodríguez-Robles et al present the results of jointly using GPR and ERT for study soil water and root dynamics. I cannot evaluate this as an expert in these geoscientific tools, but as a scientist with a general interest in water uptake and a potential end user in the long term. Considering this perspective, I found the manuscript

provides a set of novel tools to the vexed problem of assessing root water uptake, particularly in semi-arid, rocky environments. The authors show how this is the first study using this technique shallow and rocky soils, rendering the manuscript as novel. The graphics and overall data presentation is outstanding (although axes labels could be made bigger for ease of reading).

Authors: We thank the referee for his comments. In this new version we improved the font size of figures to facilitate readability.

Comment 2. I only have a couple of minor comments, in case they help to improve even more this study: 2.2 – PST 55 (notice the typo) are notorious for lack of temperature compensation. How did you account for that?

Authors: We corrected the typographical error (page 5, line 2). Regarding the lack of temperature compensation by the psychrometers, soil psychrometers (model PST-55) are designed with materials of high thermal conductivity to minimize temperature gradients. It includes a stainless-steel screen to allow only the water vapor to enter the sensor improving water vapor exchange with the soil thereby reducing internal condensation and maintaining temperature equilibrium. For this, the sensor must be buried at least to 12 cm under the soil surface. Also, to minimize thermal gradients, we installed the psychrometers with the axis of the sensor parallel to the soil surface. In our case, depending on the depth of soil pockets we installed the psychrometers between 12 - 15 cm deep. The soil psychrometer were monitored with a HR-33T microvoltimeter (Wescor) in the dewpoint mode.

Comment 3. Table 5 – please pay attention to wording of legend.

Authors: Legend of Table 5 was revised and corrected with the following text, "Nested two-way analysis of variance for examine differences in root diameter for four soil depths (10, 20, 30 and >30 cm) comparing forest stands (Pinus cembroides, Quercus potosina and Mixed) in a semiarid forest ecosystem in Central-North México." (page 19, table 5).

Comment 4. Please clarify x-axis and legend in fig. 2b.

Authors: We improved and increased font size of x-axis and clarified legend in Figure 2b as follows; (a) Relationship between root diameter from different stands and time interval with zero crossing of detected roots, which were later used for calibration with both GPR systems: 500 MHz frequency antenna (n = 48), P < 0.0001, and 800 MHz frequency antenna (n = 28), P < 0.0001. (b) Average diameter of roots recorded with GPR for each of the three forest stand types at four depths. Different letters next to the bars indicate statistical differences among treatment combinations at a probability value of P < 0.05.

Comment 5. Note also that roots are thicker at the top...Does the results of thicker roots in pine at 30 cm indicates that 1 single big root at the top divides into more thick roots afterwards?

Authors: What the figure shows is the different adaptations to rocky soils by the two forest species. Quercus in the one side, distributes a large proportion of its thicker roots on the top soil (0-10 cm depth, Fig. 2b) whereas its thin roots are located into the deepest rock fractures at the site (from 20 to more than 30 cm). For Pine, the opposite pattern of root distribution is observed. The sampling intensity used with the GPR survey did not allow us to examine root architecture as to be able to distinguish how roots divide into subsequent root orders. An additional study (Rodriguez-Robles et al., in preparation) examining the natural isotopes signature of plant tissue, soil and rock water as well as the hydrological status of plants showed that Quercus gets its water from the fractured rock whereas pine gets it from the top soil, coinciding with the distribution of fine roots.

Comment 6. How do you explain having more thick roots at deeper surfaces for pines? That's particularly surprising because, as your diagram in Fig. 3 shows, pines have a shallow root system.

Authors: As explained previously, root distribution in pine is an adaptation to a preponderance of small precipitation pulses (60% of rain events < 5mm) in shallow (< 25cm depth) and rocky soils. Thus, fine roots located at the top soil allows pine to profit from those precipitation pulses as well as soil water remobilized from the rock fractures to the top soil by oak. Thick roots have more a function of anchoring the tree.

Comment 7. Fig. 5 – there are some fairly low soil water potentials. Why?

Authors: The values of soil water potentials for this system ranged between -1 to -6 MPa and were also reported in a previous study at the site (Rodriguez-Robles et al., 2015). Values under -7 MPa, represent water potential observed in weathered volcanic rocks whereas values up to -24 MPa represent those observed in fresh bedrock. For this study, we observed that soil resistivity values ($\Omega$ Âů m-1) correlated negatively with soil water potentials ($\Psi$s, Fig. S6). The soil water potentials reported in this manuscript coincide with soil water potentials reported in other ecophysiological studies in semiarid forest in North America (McDowell et al. 2008; Hagan et al 2009).

Hagan DL, Jose S, Thetford M, Bohn K. 2009. Production physiology of three native shrubs intercropped in a young longleaf pine plantation. Agroforestry Systems 76(2): 283-294.

McDowell N, Pockman WT, Allen CD, Breshears DD, Cobb N, Kolb T, Plaut J, Sperry J, West A, Williams DG, et al. 2008. Mechanisms of plant survival and mortality during drought: why do some plants survive while others succumb to drought? New Phytologist 178(4): 719-739.

Rodriguez-Robles U, Arredondo JT, Huber-Sannwald E, Vargas R. 2015. Geoecohydrological mechanisms couple soil and leaf water dynamics and facilitate species coexistence in shallow soils of a tropical semiarid mixed forest. New Phytol 207(1): 59-69.

---

## Author Comment (AC2) · 29 May 2017

Answers to referee #2, comments on 08 May 2017.

We have reviewed and answered all comments by referee #2. We appreciate the general comments to the manuscript by the anonymous referee #2 and responded to his/her particular annotations as follow:

Comment 1. L23 in Abstract: There is a typo in the spelling of ERT.

Authors: Revised and corrected, page 1, line 23.

Comment 2. L3, page 5: please be more precise (you mean that sensors were inserted

at 12 cm depth only when allowed by bedrock?).

Authors: As noted, the terrain is very shallow and rocky, so we were referring that soil psychrometers were inserted between 12 to 15 cm depth, in available soil pockets close to our target plant. We inserted the following text instead, "which were inserted between 12 to 15 cm depth (page 5, lines 2-3).

Comment 3. L18, page 6 and elsewhere: Please consider the option of using the terms "bedrock" and "weathered bedrock" (instead of just "rock") whenever appropriate, and please replace throughout the text.

Authors: We thank the reviewer for this observation, we have considered the option of using the terms of "bedrock" and "weathered bedrock" and inserted whenever was needed in the text.

Comment 4. Line 10, page 9: leaf litter accumulates under exfoliated rocks, or on exfoliated rocks (L26, Page 10), or both?

Authors: corrected with the following text, "leaf litter accumulation under and on top of exfoliated rocks" (page 9, line 10).

Comment 5. Lines 14-17 in page 8 and again in Lines 16-18 in page 11: this appears like an unusual rooting pattern, did you find any other published papers reporting similar rooting patterns in oaks or in other tree species? Can you provide any references of similar findings?

Authors: Using stable isotope analysis of xylem water ($\delta$18O y $\delta$2H), we identified that oak trees are able to remain active during drought using alternative water sources in the substrate. Thus, oak locates their finest roots into the fractured rock, which is revealed by its particular isotopic signature (Rodriguez-Robles et al., in preparation). Also, our results suggest that use of water from the top soil by oak is limited by the presence of pine roots, likely imposing competition conditions for water (Rodriguez-Robles et al., in preparation). In another study, del Castillo et al. (2016) reported similar patterns of

root distribution for Quercus ilex and Pinus halepensis, two typical Mediterranean tree species coexisting in a mixed forest.

Comment 6. Line 27-28 in page 8: Why is this finding so surprising, am I missing something here...?

Authors: Regarding this question, our surprise arose from the fact that we did not expect that the GPR method could allow us to distinguish diameter changes along a single root. We are moderating our statement by deleting the surprising term.

Comment 7. Line 10, page 9: only leaf litter accumulated under rocks? what about root litter, was it also present under rocks? Were you able to distinguish between living and dead roots?

Authors: Using either the 500 or the 800 MHz antennas, we have not been able to identify dead roots in these types of shallow rocky soils. On the other hand, to be able to identify active roots in these shallow and rocky soils, we had to carry out GPR profiles during dry periods, when the soil is less conductive and signals from humid active roots have the highest contrast in the radargrams. In the dry periods, alive roots have a very well defined reflected hyperbolic signal, due to its water content and ions concentration. On the other hand, accumulated leaf litter under and on top of exfoliated rocks produced noisy signals in radargram traces, particularly when using the 800 MHz antenna as shown in Figure 4. We have been able to improve signal responses and interpretation using processing routines.

Comment 8. Line 7, page 12: Please clarify what exactly is meant by regolith here (weathered bedrock only? all types of bedrock?).

Authors: We refer to fragmented material of weathered bedrock observed between exfoliated rock layers and the forest floor surface. We added the following text to clarify this point "we want to highlight the major limitations encountered in this study; certain field conditions (e.g., leaf litter, weathered bedrock regolith)" page 12, lines 7-9.

Comment 9. For the sake of clarity, please rephrase item IV in lines 11-12 of page 12, this sentence is a bit confusing.

Authors: We change the text of point IV to improve clarity: "given the contact resistance problem for electrodes in the ERT survey that result especially during dry periods, from moisture content in the soil-bedrock and soil temperature." page 12, lines 12-14.

Comment 10. The quality of the figures is rather good, although I have some suggestions for improving Fig 5 (whose size in the final version should be at least twice as big as that in the PDF version that I have reviewed). I was a little confused by the legend of Fig 5, as it is not clear to me whether the three layers mentioned in the legend (soil, intermediate, bottom) are depicted or delineated in any way in the figure or not...It appears that only the water potential categories are represented by different colors.

Authors: We thank observations regarding this figure. Figure 5 has currently dimension of 1588 x 1658 pixels (300 dpi), however we have no problems to increase the size in the final version. Regarding the three substrate layers mentioned in the legend, only two; the intermediate and bottom layers represented by soil pockets-rock fractures the first and the fresh bedrock the second, are depicted in the figures. Thus, the intermediate layer is depicted with a dotted band following the GPR profile whereas the fresh rock is delimited by a solid line. The top soil corresponds to the first20-25 cm in the tomograms. Based on this comment, we have included the following text in the legend of Figure 5 "The top soil corresponds to the first 20-25 cm layer, the intermediate layer include soil pockets and rock fractures and is depicted by the dotted strip along the radargram and the fresh bedrock begins underneath the solid line" page 22, lines 4-5.

Comment 11. Please note that some of the "soil" water potential values shown in this figure (-24 MPa) are extremely and unusually low for soil and require further clarification.

Authors: The reviewer is correct, there are some fairly low soil water potentials, however these represent resistivity values for the fresh bedrock extrapolated to water potential units.

Comment 12. Applying the terms "Increasing rock moisture content" and "soil water potential" to the same moisture potential data appears rather contradictory. For the sake of clarity, I recommend to change to "Soil/bedrock water potential" and "Increasing soil/bedrock moisture content".

Authors: We adopted the suggestion by the reviewer and modified the terms in figure 5 as recommended.

Comment 13. Figure 3 should also be enlarged in the final version of the published paper. With the current size, it is very difficult to spot the B (in A) mentioned in the legend...

Authors: We thank the observations for this figure. Accordingly, we have increased the size of letters along the different radargrams (in A), for better sighting. We also followed the suggestion to increase the size of figure 3, in the final version.

Comment 14. Table 5: four soil DEPTHS....comparing forest STANDS...

Authors: Revised and corrected with the following text, "Nested two-way analysis of variance to examine root diameter differences observed among the combination of four soil depths (10, 20, 30 and >30 cm) and three forest stands (Pinus cembroides, Quercus potosina and mixed forest) in a semiarid forest ecosystem in Central-North México" (page 19, table 5).

Comment 15. The correct reference for Querejeta et al (2007) is: Querejeta JI, Estrada Medina H, Allen MF, Jimenez-Osornio JJ (2007) Water source partitioning among trees growing on shallow karst soils in a seasonally dry tropical climate. Oecologia 152:26–36.

Authors: Reference corrected.

Reference

del Castillo J, Comas C, Voltas J, Ferrio JP (2016). Dynamics of competition over water in a mixed oak-pine Mediterranean forest: spatio-temporal and physiological components. Forest Ecology and Management 382, 214-224. doi:10.1016/j.foreco.2016.10.025.

---

## Referee Report (RR1)

Overall, I want to thank the authors for drafting this manuscript entitled "Application of geophysical tools for tree root studies in forest ecosystems in complex soils". The manuscript was well written and carefully developed. Having previously conducted similar root studies, I appreciate the long, meticulous hours collecting data and analyzing the results in root systems. Root systems represent a significant uncertainty in our understanding of ecosystems and representation of plant process in model frameworks.

The paper represents a continued interest in applying geophysical techniques to ecological questions. I think the authors do a great job at presenting the problem and application of GPR to this research question, however, they don't do a great job at highlighting potential limitations of the technique. From my experience with GPR, I have a hard time believing the authors were able to identify 0.6 diameter roots in rocky soils with a low frequency antenna (e.g., 500 MHz). Past studies we were able to see slightly smaller roots with significantly higher frequency (1500 MHz) antennas and working in ideal sandy soils. I also feel the authors could have explained how they classified "noise" and how they dealt with that aspect in their methods. Additionally, they didn't clarify if the signal they were detecting was a single root or a cluster. Finally, I worry they didn't provide enough details on the settings of the antenna (e.g., signal gain points), nor did they report any calibration of the antenna/method for their sites. All GPR practitioners will say that users must do some level of calibration to understand their site to adequately decipher their radiogram results. It's not clear if they have done this or just powered on the instrument and just used the factory default settings. I would recommend they provide some additional details before this is published. Once addressed, this paper will add to the growing knowledge of how to study root systems.

Introduction

The introduction was well written and presented a great overview of the topic. Unfortunately, the introduction lacked any mention of the real tradeoffs of using GPR in ecological issues. For example, there is a strong tradeoff between antenna frequency, resolution and depth. Higher frequency antennas (e.g., 2,000 and 1,500 MHz) provide greater ability to resolve smaller roots, but lack depth penetration (~0.5 meters), whereas a lower frequency antenna (e.g., 400 MHz), will provide increased depth, but only be able to detect larger root structures. Additionally, the authors don't adequately provide any details on the issues of overshadowing. Larger roots and rocks near the surface will inhibit the ability to discern high resolution details of smaller roots unless you have large transects where the (cone shaped)signal reflectance is interpreted from away, on top of, and past the target. I was disappointed the introduction lacked any clear description of the types of tree at the SSM or their typical root characteristics (e.g., tap root). Finally, the second objective "with the use of GPR, detecting roots of different diameter size classes growing at various depths in volcanic fractured rock" – Why just determine diameter? You could easily estimate root mass or architecture from this study and would likely be the most useful product for other studies/models to use (diameter is a poor indicator of age, mass, etc.).

Methods

Why were the plots circular? Circular data collection leads to some problems with interpreting such issues. There have been some companies (e.g., Tree Radar), that had to develop new methods to

analyze tree trunk decay from GPR images collected around the trunk (in other words, in a linear transect, you can compare reflector intercepts, but in a circular scan pattern, you will have to deal with adjacent intercepts that will be dealing with different intercept angles at the edges). How did you account for this in your analysis? However, with a circular pattern, you could determine if there is any overshadowing of rocks or other roots.

Just using 0.1, 0.3, 0.5, 1.0 and 1.5 m transects – its unlikely you could detect small diameter (~0.6cm) roots with a 500 MHz antenna or a 900 MHz antenna with such large gaps (especially with the higher frequency antenna).

Did the soil sensors have any metal in them? If so, how did you deal with issues with signal attenuation?

On Page 5, when you talk about the principles of GPR, you could mention the resolution-depth-frequency tradeoff, but it might be better in the introduction.

What about air gaps? Wouldn't volcanic rock have a large number of "air bubble" gaps? How would you deal with issues associated with air gaps? From my experience, this causes change in the signal speed and can cause issues with your gain settings and therefore interpretation.

In page 6, section 2.5, to what depth?

How did you deal with the immediate ground reflection from the antenna? Were there any gaps (due to micro topography? If so how did you deal with this?

Page 6, line 26-31, over the 8 month sampling period, did you collect the ERT at the same time as the GPR? Since soil water content can change so rapidly in arid/semi arid regions, this would be important especially since changing moisture levels will impact your dielectric constant.

The manuscript makes no mention of site calibration for your equipment/method. From my experience, you need to calibrate GPR to the site conditions, especially with respect to signal gain points. It was unclear how many gain points were used and if they changes between plots (using the "automatic gain detection feature often isn't the best for mid to high resolution studies where you want to compare between plots.

Page 7, line 6 – how did you determine unwanted signal (noise)? Any criteria? This would be useful for readers to understand this step in case they wanted to use this methodology. I would suggest looking at or referencing methods listed in the book "Measuring Roots: An Updated Approach (Springer; Editor: S. Mancuso - ISBN 978-3-642-22067-8) or "Handbook of Agricultural Geophysics" (CRC Press; Editors: B> Allred, J. Daniels, M.R. Eshani - ISBN 9780849337284).

Did you do any ground truthing of this method? This would be important to know what depths you are really reaching with this method. Some past studies have inserted a metal reflector pate at a known depth (e.g., 50cm) to ensure proper calibration of the data. Otherwise you are just making assumptions of how deep you are penetrating. I realize you did dig up some surface roots for a comparison, but differences in the soil, air gaps, variations in soil moisture, etc. can speed up or retard the signal travel time resulting in changes in depth interpretations.

Why not use a higher frequency antenna like 1500 or 2000 MHz since most of your roots are in the upper 50 cm of the soil?

Soils and root systems are highly variable and roots tend to criss cross/overlap areas in the soil, how did you partition roots out in the radar where they overlap (or grow side by side) and not consider them as 1 root, rather than 2 smaller roots). With such a low freq. antenna your diameters could be smaller roots in a group, rather than one 2.5 cm diameter root.

Results

Page 8, line 26 – The authors suggest you could track elongation over time, but I doubt you could detect this with the sampling frequency and the reported diameter sizes, unless these are fast growing trees.

Its' unclear how the authors dealt with the shadowing from rock fragments and potential gaps or microsites of moisture in a crevice in their radiograms?

Page 9, line 11, its unclear what the authors mean by "spotted primarily" tree roots – what else was detected? this leaves some doubt in the detection analysis.

How did the authors tease out species specific information? (e.g., Page 9, lines 13-19)? Were these monoculture patches of species?

Page 9, line 5, wouldn't you need less than a meter for the 500 MHz to detect 0.6 cm roots? the 900 would likely detect this size range

Page 9, line 24-25 – could this finding be due to the differences in rooting strategies in Oaks and Pine species?

Page 10, line 1 – how long did the water infiltration signal last? Also, when was this objective/phenomena studied? This could matter because the transpiration and plant water demands could change the interpretation of the duration of the perturbation.

Discussion

Page 10, line 22-23, I agree with the dual frequency approach given differences in sites and a method I would use (the art of the method), but again, you need to do custom your method to your site (ground truthing/validation).

Table 1 – I doubt they are seeing 0.6 cm diameter roots with the lower frequency antennas in this rocky soil substrate. Past studies have achieve this only with 1500-2000 MHz antenna only in ideal soil settings (sandy soils)

Table 4 – the authors say "used for calibration" but don't really explain what they mean here? Also, It would be useful to show those regressions here.

Figure 1 – the rockiness of the soil, and possible air gaps, would make signal processing very difficult here, each plot would need to be calibrated.

Figure 2 – what's the age class of the trees?

Figure 3 – please italics the species names

Figure 4 – looks like you have some attenuation of the signal (e.g., where label "a" is located). Also what do the lowercase letters represent?

---

## Author Response (AR2)

**Response to comments RC3**

We have now reviewed the comments by RC3. We appreciate the suggestions and comments posed by the referee. Below we include the answers to all the questions and they were incorporated into the manuscript.

Please find below the response to each of their comments in *blue italics.*

**Comment 1**. The paper represents a continued interest in applying geophysical techniques to ecological questions. I think the authors do a great job at presenting the problem and application of GPR to this research question, however, they don't do a great job at highlighting potential limitations of the technique.

*Authors: **In the version reviewed by** We had already addressed the major limitations encountered in this study inserted in the conclusion section (pages: 12 and 13, lines: 31-33 and 1-3). Following the comment, we included some lines in the introduction section, highlighting potential limitations of the technique:*

*"However, various factors affect detection of roots using GPR, such as root position, wood density and the conditions surrounding roots. These conditions include for instance, physical properties, altered or removed material, the volumetric water content, temperature, dissolved solids or salinity, the existence of regolitic material, and applied GPR wave frequency. These conditions may interfere with signal transmission and thus resulting in low-quality and difficult-to-interpret profiles (Table 1). For example, root zones in wet conductive soils, high-frequency waves are strongly attenuated limiting the resolution to detect roots and depth penetration (Butnor et al., 2012)" (page 3, lines 14-19).*

**Comment 2.** I also feel the authors could have explained how they classified "noise" and how they dealt with that aspect in their methods.

*Authors: We implemented the "Bandpass filtering routine" to eliminate much of the noise in the radargrams. Noise seriously affects the judgement of reflective surfaces. However, noise can be removed by vertical high-pass and low-pass filtering. A high-pass filter, removes the low frequency signal content in the temporal dimension. It is used to remove*

*low frequency data and horizontal banding due to system noise. Based on this comment, we extended the description of the filtering method of the signal-to-noise ratio following the steps described Butnor, J. R., et al. (2012) chapter. (pages: 7 and 8, lines 15-33 and 1-6). (pages: 7 and 8, lines 15-33 and 1-6).*

**Comment 3**. Finally, I worry they didn't provide enough details on the settings of the antenna (e.g., signal gain points), nor did they report any calibration of the antenna/method for their sites. All GPR practitioners will say that users must do some level of calibration to understand their site to adequately decipher their radiogram results. It's not clear if they have done this or just powered on the instrument and just used the factory default settings. I would recommend they provide some additional details before this is published.

*Authors: As suggested by the reviewer we have replaced the section "Data processing" by "Data processing and interpretation". The configuration to remove the DC component to offset the signal gain, and also were included the adjustment of other equipment parameters (pages: 7 and 8, lines 15-33 and 1-6). Regarding the calibration and validation of radargrams, we used in-situ roots for calibration (76 roots total). In Figure 2a we present the relationship between root diameter and signal emitted by the GPR system for all experimental plots. Also, in Table 3, we present values of roots measured in-situ as well the corresponding information from the GPR signal that we used for calibration and validation of roots (diameter and depth). The Table 3 represent a radargram of a pine-oak plot (Fig. S4).*

- ***Introduction***

**Comment 4**. The introduction was well written and presented a great overview of the topic. Unfortunately, the introduction lacked any mention of the real tradeoffs of using GPR in ecological issues.

*Authors: This comment was addressed on lines (Page 3, lines 14-19). The text describing these limitations is also shown in comment 1 above.*

**Comment 5**. I was disappointed the introduction lacked any clear description of the types of tree at the SSM or their typical root characteristics (e.g., tap root).

*Authors: Little is known about root morphology and distribution of these two particular species growing on shallow rocky soils. In page 4, lines 2-3, we added the following text addressing the reviewer's concern: "overall Quercus species exhibit dimorphic roots, while Pinus species display preferently superficial roots spreading horizontally (Kutschera & Lichtenegger, 2002)."*

**Comment 6**. Finally, the second objective "with the use of GPR, detecting roots of different diameter size classes growing at various depths in volcanic fractured rock" – Why just determine diameter? You could easily estimate root mass or architecture from this study and would likely be the most useful product for other studies/models to use (diameter is a poor indicator of age, mass, etc.).

*Authors: We thank the reviewer for this recommendation. However, we implemented geophysical tools in our studies, as a need to answer a research question about the operation of water redistribution (hydraulic lift) n this semi-arid rocky forest with shallow soil, question that arose from a previous study Rodriguez-Robles, U., Arredondo, J. T., Huber-Sannwald, E., and Vargas, R.: Geoecohydrological mechanisms couple soil and leaf water dynamics and facilitate species coexistence in shallow soils of a tropical semiarid mixed forest, The New phytologist, 207, 59-69, 10.1111/nph.13344, 2015.*

*As such, we prepared the present manuscript only with the root diameter information we recorded to address the mentioned question.*

- *Methods*

**Comment 7**. Why were the plots circular? Circular data collection leads to some problems with interpreting such issues. There have been some companies (e.g., Tree Radar), that had to develop new methods to analyze tree trunk decay from GPR images collected around the trunk (in other words, in a linear transect, you can compare reflector intercepts, but in a circular scan pattern, you will have to deal with adjacent intercepts that will be dealing with different intercept angles at the edges). How did you account for this in your analysis?

However, with a circular pattern, you could determine if there is any overshadowing of rocks or other roots.

*Authors: For the present study, we used three different transect types to carry out the GPR tests for root survey. First, we implemented a radial transect oriented in four directions (NS, NE-SW, EW and SE-NW) within each experimental plot of 25 m diameter, using a 500 MHz antenna for the 12 experimental plots. Another transect consisted of the concentric sampling mentioned by the referee, each sampling included five concentric transects (0.1, 0.3, 0.5, 1.0 and 1.5 m between neighboring transects) around a Pinus cembroides tree anchored within an exfoliated rock using the 800 MHz antenna. The objective of surveys following concentric transects was to figure it out whether Pine located roots underneath the exfoliated rock as a source of humidity and nutrients as well as to observe how roots distributed roots away from the rock. In this case, we only worked with reflected hyperbolas that result from perpendicular roots to the GPR direction. Therefore we only used well defined reflected hyperbolas. Finally, the third transect type was established inside 8.5 x 6 m plots, consisting in seven parallel transects (spaced 1 meter) to observe horizontal axis elongation of roots in a mixed stand.*

*This experimental arrangement was addressed in the section "2.2 Experimental plots" page 5, lines 4-79.*

**Comment 8**. Just using 0.1, 0.3, 0.5, 1.0 and 1.5 m transects – its unlikely you could detect small diameter (~0.6cm) roots with a 500 MHz antenna or a 900 MHz antenna with such large gaps (especially with the higher frequency antenna).

*Authors: In transects of 1.0 and 1.5 m, we detected small diameter roots (0.6 cm) using the 800 MHz antenna. We validated with direct measurements of roots in situ the hyperbolic signal, to calibrate the radargrams.*

**Comment 9**. Did the soil sensors have any metal in them? If so, how did you deal with issues with signal attenuation?

*Authors: We understand the concern of the reviewer. The soil psychrometer sensors (PST-55), they are designed with high thermal conductivity materials. we installed the psychrometers with the axis of the sensor parallel to the soil surface and depending on the*

*depth of soil pockets we installed the psychrometers of 12 - 15 cm deep. Before the surveys with GPR and ERT, we performed the measurement of the water potential and after that, the sensors were extracted for the geophysical surveys.*

**Comment 10**. On Page 5, when you talk about the principles of GPR, you could mention the resolution-depth-frequency tradeoff, but it might be better in the introduction.

*Authors: We took into account the reviewer's comments; the section was modified as follows: "The ground electrical conductivity, the transmitted center frequency, and the radiated power, all may limit the effective depth range of GPR prospection. Increases in electrical conductivity in the ground may attenuate the introduced electromagnetic wave, and thus its penetration depth decreases. However, higher GPR frequencies may provide improved resolution. Hence, operating frequency is always a trade-off between resolution and ground penetration (Aditama et al., 2017)." Page 3, lines 1-5.*

**Comment 11**. What about air gaps? Wouldn't volcanic rock have a large number of "air bubble" gaps? How would you deal with issues associated with air gaps? From my experience, this causes change in the signal speed and can cause issues with your gain settings and therefore interpretation.

*Authors: As mentioned in the methods, the study was carried out in the SSMVC that represents the remnants of one of the most voluminous rhyolitic volcanic events on Earth (McDowell and Keizer, 1977), formed by massive lava spills of rhyolitic composition (Portezuelo Latite and San Miguelito Rhyolite). Rhyolite is an extrusive igneous rock, making it impermeable and void of porosity. The weathering of this rock is by lapping, so there are no abrupt changes in the rock surface. Still, to minimize issues associated with air gaps, we set the "set zero level" to compensate for signal gain.  In addition, the "auto stacking" was configured for highest data quality and optimized speed performance.*

*To run the root surveys, we first configured the antennas [time windows (400 ns), stacks (auto), sampling frequency (1120 MHz), point distance (0.001 m) and velocity (110 m/µs)] and used a pulling system and a measuring wheel fitted to the mounting block on the back of the shielded antenna. The antenna was dragged along the transect trying to always maintain it in close contact with the soil.*

**Comment 12**. In page 6, section 2.5, to what depth?

*Authors: We added the depths as shown in the following text; "From October to December 2012, we examined the frequency, size, position and depth range of roots with 2.5 to 7.5 cm diameter in the top soil and under exfoliated rocks, using the GPR 500 MHz antenna and working with a 2.5 m depth window. To characterize the exfoliation of weathered bedrock soil, and to differentiate between the exfoliated rock base and potential root axes (0.6 to 4 cm diameter) underneath rocks, we used the GPR 800 MHz antenna, exploring at 1 m depth". (Page 6, lines 25-28).*

**Comment 13**. How did you deal with the immediate ground reflection from the antenna? Were there any gaps (due to micro topography? If so how did you deal with this?

*Authors: To deal with this problem, we focused on signal processing from a reduced clutter that better represent the geometry of roots (as mentioned in response to Comment 2, addressed in section 2.6). It is worth noting that we carried out the root surveys using MALA shielded antennas (consists of both transmitter and receiver antenna elements in a single housing) obtaining signals and reflection data that are cleaner for interpretation.*

**Comment 14**. Page 6, line 26-31, over the 8month sampling period, did you collect the ERT at the same time as the GPR? Since soil water content can change so rapidly in arid/semi arid regions, this would be important especially since changing moisture levels will impact your dielectric constant.

*Authors: Thanks for the observation, the GPR surveys were carried out in October 2012 during a drought period (45 days of soil drying, previous to the survey) to assure cleaner signals and reflectance of roots in these complex soils. ERT profiles in contrast, were carried out one year later, beginning in a wet period (October 2013) and lasting for eight months until the dry period (May 2014), with the objective as defined for this experiment to "determine if resistivity tomography help to detect the spatial and temporal variability of soil moisture beneath vegetation patches" and to "describe the role of weathered bedrock in forest ecosystems colonizing shallow rocky soils.*

*With this arrangement we were able to overlap GPR and ERT profiles and show the relationship between root location, low soil resistivity (greater water availability) and greater bedrock fracturing, as shown in the figures 5 and S5.*

**Comment 15.** The manuscript makes no mention of site calibration for your equipment/method. From my experience, you need to calibrate GPR to the site conditions, especially with respect to signal gain points. It was unclear how many gain points were used and if they changes between plots (using the "automatic gain detection feature often isn't the best for mid to high resolution studies where you want to compare between plots).

*Authors: In section 2.6 it is mentioned both the calibration method as well as the radargam processing using RadExplorer software (page 7: lines 15-33 and page 8: lines 1-6). To calibrate and validate the root survey method we proceeded as follows; "Root identification in the radargrams was a stepwise process; first, roots were recognized at the locations where hyperbolas of reflected waves had higher amplitudes compared to those in the surrounding area (Cui et al., 2011). Then, to determine the diameter and depth of those roots, the time interval between zero crossings (ns, time interval for maximum reflected wave) was extracted at the points of hyperbolas, where roots had been identified previously. The detection frequency for the number of roots identified in the radar profile was calculated along each transect for five root diameter classes (< 3.0, 3.0 – 4.0, 4.0 – 5.0, 5.0 – 6.0, and > 6.0 cm). Finally, for calibration purposes, individual roots (total of 76) were excavated to determine their depth and diameter in-situ (Table 3, Fig. 2a). (Section 2.5, pages: 6 and 7, lines: 29-31 and 1-4).*

**Comment 16**. Page 7, line 6 – how did you determine unwanted signal (noise)? Any criteria? This would be useful for readers to understand this step in case they wanted to use this methodology. I would suggest looking at or referencing methods listed in the book "Measuring Roots: An Updated Approach (Springer; Editor: S. Mancuso - ISBN 978-3-642-22067-8) or "Handbook of Agricultural Geophysics" (CRC Press; Editors: B> Allred, J. Daniels, M.R. Eshani - ISBN 9780849337284).

*Authors: Answer to this question was already addressed in comment 2. However, based on this comment, we have reorganized the section as indicated by the reviewer and also included the references for the signal-to-noise ratio issue described in the chapter of*

*Butnor, J. R., et al. (2012). Using Ground-Penetrating Radar to Detect Tree Roots and Estimate Biomass. Measuring Roots: An Updated Approach. S. Mancuso. Berlin, Heidelberg, Springer Berlin Heidelberg: 213-245. (pages 7 and 8, lines 15-33 and lines 1-6).*

**Comment 17**. Did you do any ground truthing of this method? This would be important to know what depths you are really reaching with this method. Some past studies have inserted a metal reflector pate at a known depth (e.g., 50cm) to ensure proper calibration of the data. Otherwise you are just making assumptions of how deep you are penetrating. I realize you did dig up some surface roots for a comparison, but differences in the soil, air gaps, variations in soil moisture, etc. can speed up or retard the signal travel time resulting in changes in depth interpretations.

*Authors:  For calibration purposes, individual roots (total of 76) were excavated to determine their depth and diameter in-situ (Table 3, Fig. 2a). Using the "Tools / Depth Calibration" tool of the ProEx system, in-situ, we calibrated the depth of GPR radargam with that obtained by the excavation.*

**Comment 18**. Why not use a higher frequency antenna like 1500 or 2000 MHz since most of your roots are in the upper 50 cm of the soil?

*Authors: Limited by ground conditions (e.g., leaf litter, weathered bedrock regolith), we decided to work with the 500 and 800 MHz antenna. Following a rigorous processing in the Rad Explorer, we had obtained good results with the 800 MHz antenna for these floors with high presence of needles. Also, the use of 1500 or 2000 MHz antennas introduces a lot of noise coming from superficial debris and roots from the understory vegetation.*

**Comment 19**. Soils and root systems are highly variable and roots tend to crises cross/overlap areas in the soil, how did you partition roots out in the radar where they overlap (or grow side by side) and not consider them as 1 root, rather than 2 smaller roots). With such a low freq. antenna your diameters could be smaller roots in a group, rather than one 2.5 cm diameter root.

*Authors: To solve the problem due to the hyperbolas overlapping as well to identify the correct position of roots, we applied a Stolt F-K Migration routine. With the application of*

*the Stolt F-K Migration routine we were able to restore the real location and shape of reflecting boundaries in a section plane. On the other hand, to determine the wave propagation velocity, we relied on the "Hyperbola" tool (Rad Explorer software) in order to determine migration velocity.*

*Thus, after the Stolt F-K Migration routine was applied, if resulted reflections were not clear to interpret, that section was omitted.*

- ***Results***

**Comment 20**. Page 8, line 26 – The authors suggest you could track elongation over time, but I doubt you could detect this with the sampling frequency and the reported diameter sizes, unless these are fast growing trees.

*Authors: The reviewer is correct, what we describe in this lines is the horizontal distribution of a root rather than its elongation. With the 500 MHz antenna, we detected the horizontal distribution of a root axis of Pinus cembroides in a 8.5 x 6 m plot running parallel transects (Figure 3 B). Through in situ excavations we were able to validate the hyperbolas reflected in the radargrams.*

**Comment 21**. Its' unclear how the authors dealt with the shadowing from rock fragments and potential gaps or microsites of moisture in a crevice in their radiograms?

*Authors: The GPR surveys were carried out during a period of drought (October 2012), to minimize signal noise in the radargrams and facilitate their interception. However, a limitation of using the GPR technology, relates to the interpretation of GPR images. To reduce this problem and when the conditions at the site allowed, following a GPR survey we proceeded validation through excavations removing soil and removable rocks as shown in the following figure.*

[Figure]

*In the image is observed (after soil and rock remotion), a pair of roots of Pinus cembroides tree (anchored to a volcanic rock) that served to calibrate radargrams for the radial transects.*

**Comment 22**. Page 9, line 11, its unclear what the authors mean by "spotted primarily" tree roots – what else was detected? this leaves some doubt in the detection analysis.

*Authors: Regarding the observation of reviewer, we refereed to the noisy signal generated by leaf litter accumulation under and on top of exfoliated rocks.*

**Comment 23**. How did the authors tease out species specific information? (e.g., Page 9, lines 13-19)? Were these monoculture patches of species?

*Authors: The reviewer is correct, the experimental study was set up in a pine-oak forest. Along a 2.5 km long transect we established a total of 12 circular experimental plots of 25 m diameter with four replicates for three types of forest stand types (pine, oak, and mixed stands). Page 5, lines 4-9.*

**Comment 24**. Page 9, line 5, wouldn't you need less than a meter for the 500 MHz to detect 0.6 cm roots? the 900 would likely detect this size range.

*Authors: Roots smaller than 1cm in diameter were only detected with the 800 MHz antenna in our case.*

**Comment 25**. Page 9, line 24-25 – could this finding be due to the differences in rooting strategies in Oaks and Pine species?

*Authors: The reviewer is correct, we assume that Pinus cembroides have a more limited geospatial niche than Quercus potosina and this is due to differences in rooting strategies between Oaks and Pine.*

**Comment 26**. Page 10, line 1 – how long did the water infiltration signal last? Also, when was this objective/phenomena studied? This could matter because the transpiration and plant water demands could change the interpretation of the duration of the perturbation.

*Authors: This was an additional test only to demonstrate the potential of the geophysical methods to detect short-term horizontal and vertical distribution of water in the substrate (e.g., in response to a rain pulse). We have re-worked the objective 4 (page 4, lines 17-18). Radargrams showed both a clear infiltration horizon at approximately 50 cm depth and a remarkably rapid horizontal displacement of this injected water. The signal appeared only 150 minutes following water injection and it was not homogeneous among all vegetation patches.*

- *Discussion*

**Comment 27**. Page 10, line 22-23, I agree with the dual frequency approach given differences in sites and a method I would use (the art of the method), but again, you need to do custom your method to your site (ground truthing/validation).

*Authors: Regarding this reviewer comment, we can comment that we made calibration curves on each stand type, using root diameter (cm) and time interval (ns), as shown in table 4 and figure 2a. We were very careful in the calibration and validation of the method for each stand, following our configuration method.*

- ***Tables and figures***

**Comment 28**. Table 1 – I doubt they are seeing 0.6 cm diameter roots with the lower frequency antennas in this rocky soil substrate. Past studies have achieve this only with 1500-2000 MHz antenna only in ideal soil settings (sandy soils).

*Authors: We observed hyperbolic reflectances for roots smaller than 1cm using the 800 MHZ antenna, in the concentric transect arrangement (figure 4b) transect at 150 cm (letter "b").*

*Following the filtering process presented in section 2.6 (pages 7 and 8, lines 17-33 and 1-6), we were able to eliminate and greatly improve the noise in the radargrams to facilitate interpretation. An example is shown underneath: a) Raw radargram; b) DC removal and correction of time zero c) Background removal and Bandpass filtering.*

[Figure]

[Figure]

[Figure]

**Comment 29**. Table 4 – the authors say "used for calibration" but don't really explain what they mean here? Also, It would be useful to show those regressions here.

*Authors: We thank the reviewer for this observation. Overall calibration method was explain in comment 15. Additionaly, direct measurements of root diameters were obtained from 4 different profiles. From each profile, we also conducted excavations of at least 4 roots in situ. For calibration for root diameter studies when applying the 500 MHz antenna we used the 12 experimental plots, whereas when surveying with the 800 MHz antenna we used the five concentric profiles for lateral root distribution underneath exfoliated rocks. For each of the profiles (500 and 800 MHz), roots were excavated and directly measured. In Figures 3, 4 and S4, roots used for GPR calibration are indicated with letters. Regressions were carried out with in-situ roots at different depths with those detected in the profile.*

*Table 4 including parameters of linear regression was extended.*

| Type of sampling | GPR systems | Stand | Intercept ± 1SE | Slope ± 1SE | R² | P |
|---|---|---|---|---|---|---|
| Radial | 500 MHz | Pine | 0.3610 ± 0.3751 | 5.7890 ±0.4473 | 0.92 | <0.0001 |
| Radial | 500 MHz | Oak | -0.0013 ± 0.4234 | 5.7736 ± 0.4309 | 0.92 | <0.0001 |
| Radial | 500 MHz | Mixed | -0.0536 ± 0.1506 | 6.0195 ± 0.1712 | 0.98 | <0.0001 |
| Concentric | 800 MHz | Pine | -2.0910 ±0.1273 | 6.2450 ±0.1839 | 0.98 | <0.0001 |

**Comment 30**. Figure 1 – the rockiness of the soil, and possible air gaps, would make signal processing very difficult here, each plot would need to be calibrated.

*Authors: Answer to this question is similar to that posed for comment 28.*

**Comment 31**. Figure 2 – what's the age class of the trees?

*Authors: In table 2 we have included a column with the estimated age of trees based on ring growth.*

| Stand | n | DBH (cm) | Age (years) | Tree height (m) |
|---|---|---|---|---|
| Pine/pure | 16 | 18.701 ±2.49 | 76.05 ±3.42 | 4.863 ±0.74 |
| Oak/pure | 16 | 21.104 ±1.67 | 83.17 ±3.21 | 5.272 ±0.86 |
| Pine/mixed | 16 | 19.981 ±1.76 | 84.20 ±4.88 | 6.080 ±1.17 |
| Oak/mixed | 16 | 20.121 ±1.38 | 82.06 ±2.82 | 5.461 ±1.08 |

**Comment 32**. Figure 3 – please italics the species names

*Authors: Revised and corrected*

**Comment 33**. Figure 4 – looks like you have some attenuation of the signal (e.g., where label "a" is located). Also what do the lowercase letters represent?

*Authors: The reviewer is correct, in the last two radargrams of figure 4 it is shown an attenuation in the hyperbolic reflectances, afterwrda to have applied the Bandpass filtering routine. This radargram has been largely eliminated from the noise generated by the needles present at the site. It should be noted that these radargrams were performed using the 800 MHz GPR antenna (shown in the figures, from the response of comment 28).*

References:

[revised manuscript text omitted]

---

## Author Response (AR3)

Dr. Anja Rammig
Associate Editor

Dear Dr. Rammig

We have now reviewed and answered the comments posed by reviewer 3 and by yourself. Please find below the responses to each of their comments in *blue italics.*

- **Response to comments Editor**

**Comment 1.** - It seems that the paper is more focused on method development and an initial application. Could you consider calling this a method development in the very beginning?

*Authors: It is, in fact we submitted the manuscript as a Technical note. We introduced the text of method development in the summary section (Page 1, line 20).*

**Comment 2.** - How are roots under rocks imaged, do you detect the root on both sides of the overarching rock and infer its connection?

*Authors: Answer to this question is similar to that posed for comment 24. The GPR surveys were carried out during the dry ecohydrological period (Oct-Dec 2012) to minimize signal noise in the radargrams and facilitate root signal detection. Following a GPR survey and on-site interpretation of radargrams, we proceeded root validation through excavation, removing soil and removable rocks. The RAMAC XV Monitor for ProEx system, allowed us to visualize in situ, the position, size and depth of the root.*

**Comment 3.** - How do you separate other root functions besides water uptake when you simply connect root diameter with water uptake?

*Authors: In this manuscript we do not make inferences about root functions. We just suspect that the smallest root diameter located into the humid substrate sectors should participate in root water uptake. The manuscript is focused on demonstrating the potential use of these geophysical methods.*

- **Response to comments RC3**

*Abstract*

**Comment 4.** Page 1, Line 22 – you report diameters in mm, but throughout the paper you use cm, consistency would be best to reduce any confusion.

*Authors: Thanks for the observation, we referred throughout the manuscript just to cm for consistency.*

*Introduction*

**Comment 5.** Page 3, Line 25 – What is meant by "root adaptations", or how you could determine adaptations via geophysical techniques.

*Authors: We referred to any aspect related to distribution or diameter that could relate to acquisition of substrate resources. In this study for instance; placement of roots with different diameters to different soil depth profiles (fine-deep roots may function for water uptake since its distribution corresponded to water location in the substrate).*

**Comment 6.** Page 4, Line 4 – "This study responds to a cross-disciplinary call for the application and wide use of new geophysical methods…" There have been a fair number of publications that have employed GPR to look at belowground systems and root ecology. You might change this to "developing" or "advancing" rather than new.

*Authors: As suggested by the reviewer we have changed the term "new" by "developing". "This study responded to a cross-disciplinary call for the application and wide use of other field's technology by developing geophysical methods to advance in-situ research in root ecology" (Page 4, lines 4-5).*

**Comment 7.** Page 4, Line 10 – "We expected weathered…." Shouldn't this be "we hypothesized"?

*Authors: The reviewer is correct, we switched expected by hypothesized in the Page 4, lines 9-11.*

*Methods*

**Comment 8.** Page 5, Line 6-7 – The authors report using one 8.5x6m plot with multiple transects. This appears to be very low replication and wouldn't shed light on the heterogeneity of this systems. Also does it provide enough statistical power?

*Authors: The objective of sampling this plot was to examine the method capabilities to follow root distribution on particular areas. It was not attempted to characterized root distribution at a larger scale nor run a particular comparative study. Still, with the 8.5 x 6 m plot, we obtained satisfactory results to detect the horizontal distribution of a root axis of Pinus cembroides running parallel transects (Figure 3B) with the 500 MHz shielded antenna. With in situ excavations we were able to validate the hyperbolas reflected in the radargrams. In these samples we performed the calibration with the values of the linear regression of the mixed radial plots (Table 4).*

**Comment 9.** Page 7, Line 22 – What is a "star point"?

*Authors: The "start point" is a time gain filter that compensates for amplitude reduction with depth. This parameter is part of the boot configurations of Monitor for ProEx, a priori to the measurements.*

**Comment 10.** Page 7, Line 27 – The authors mention the study area had similar characteristics, but it's unclear when the measurements were taken. The methods indicate Oct-Dec, but it's unclear if there were multiple data collection point (monthly) for each plot or once with each plot taken at different time over the three month period. If there were any environmental changes, this could impact the way you interpret the results.

*Authors: The time period mentioned corresponds for our site to a dry ecohydrological period (Oct-Dec 2012) The GPR surveys using a 500 MHz shielded antenna was carried out October 2012 in 12 circular plots of 25 m diameter with four replicates per stand type . In addition we run a survey in November 2012, in one 8.5 x 6 m plot with parallel transects to observe horizontal root axis distribution in a mixed stand. We also examined one concentric plot including five circular transects around an anchored Pinus cembroides tree (December 2012). The major environmental change at the site that could alter root data interpretation would occur with rain, however in this period we did not record precipitation at the site. We thrust data interpretation within ecohydrological periods does not change for root ecology studies.*

**Comment 11.** Page 7-8, Lines 29-6 – Since much of this paper is focused on the method development, I feel it's important to include this information. However, the way it is currently presented (as a bulleted list) made this section difficult to read and didn't flow well with the rest of Section 2.6.

*Authors: To facilitate reading of this section 2.6, we present now the processing routines unbulleted (Page 7 and 8; lines 30-33 and 1-6 respectively).*

*Results*

**Comment 12.** Page 8, Line 30-31 – "Nevertheless, radargrams indicated clear hyperbolic reflections that corresponded to the position of the tree roots at certain depths" – You should clarify this was confirmed with excavations.

*Authors: We appreciate the suggestion by the reviewer, we have emphasized in the text that detection of roots in the radargrams was validated through precise field excavations (Page 8, line 31).*

**Comment 13.** Page 9 – The authors might have mentioned this before, but what is the average depth to bedrock in this system?

*Authors: The bedrock is detected to start around 75 cm depth (views through geological sections). In the radargrams of Figure 4 it is shown the base of exfoliated rocks (physical weathering) about 35 cm deep.*

**Comment 14.** Page 9, Line 29 – The authors indicate they have determined presence of roots underneath bedrock – I question how they can detect this with the GPR signal since the major reflection of the rock would prohibit and mask this detection. I assume they are just connecting a root signal on both sides of a rock to be the same root, but it could represent different roots. The following line (31) they mention the increase of hyperbolas under the bedrock. Maybe they aren't being clear, but how are they detecting anything under a major rock formation?

*Authors: We understand the reviewer's concern. The figure 4, like all the figures that show images of geophysical prospecting, were calibrated and validated with in situ excavations. One advantage at our site is that our lived roots are embebed into a relatively dry rocky substrate, therefore reflection signals from both components can be discerned. Also, through excavations underneath exfoliated rock, we could validate the presence of roots. Following the filtering process presented in section 2.6 (Page 7 and 8, lines 16-33 and 1-6), we were able to eliminate and greatly improve the noise in the radargrams to facilitate interpretation.*

**Comment 15.** Page 10, Line 27 – What do the authors mean by "Remarkably rapid"? This is very qualitative and hard to determine the usefulness of this observation.

*Authors: Revised and corrected as. "we could detect a signal for horizontal displacement of water up to three meters distance from where it was injected" (Page 10, lines 27-28).*

*Conclusion*

**Comment 16.** Page 13, Line 5-6 – This sentence is more methodological and should be moved to the Methods section since it is important in the development and interpretation of the results.

*Authors: We thank the reviewer for this observation. We believe that the emphasis on problem solving should remain in the study's conclusions (Page 11, lines 7-8). However it is also important to mention this in the methods section. "To minimize radargrams signal noise we proceed pre-cleaning the substrate surface from litter and twigs in all experimental plots (Page5, lines 9-10 and 13).*

**Comment 17.** Table 4 – The authors should double check their R squared values, it's unusual that they are exactly the same (possible typo?)

*Authors: The reviewer is correct. We revised and corrected.*

**Comment 18.** Figure 3 – It's hard to envision how the concentric measurements were made in the circular plots from this figure.

*Authors: The following figure shows the three types of GPR surveys we carried out (radial, concentric and linear transects (8.5 x 6m)).*

[Figure]

*First, we implemented a radial transect oriented in four directions (NS, NE-SW, EW and SE-NW) within each experimental plot of 25 m diameter, using a 500 MHz shielded antenna. Another transect consisted in the concentric sampling mentioned by the referee, each sampling included five concentric transects (0.1, 0.3, 0.5, 1.0 and 1.5 m between neighboring transects) around a Pinus cembroides tree anchored within an exfoliated rock using the 800 MHz shielded antenna. The third transect type was established inside 8.5 x 6 m plots, consisting in seven parallel transects (spaced 1 meter) to observe horizontal axis elongation of roots in a mixed stand.*

Authors Responses

**Comment 19.** Comment 3, 17, 19 – These comments really just defer to using the analysis software which makes me wonder if the authors really understand how these filters really work and impact their radargrams. You really can't substitute calibration with a software correction and could lead to blind faith in software!

*Authors: The referee is correct however our intention was not substitute software routines by in situ calibration. We did apply processing routines and filters in the interpretation of software radargrams with RadExplorer v1.42 software (Mala GeoScience, USA Inc) to help us to reduce clutter, minimize the effects of multiple hyperbolic reflections and enhance the signal-to-noise ratio, removing unwanted signals (noise) through filtering of radar data and corrected the position of reflectors on the radar record. For calibration*

*purposes, A total of 76 individual roots were excavated to determine their depth and diameter in-situ (Table 3, Fig. 2a).*

**Comment 20.** Comment 6 – I understand the goal was to examine hydraulic function in this system, but I'm concerned the authors are confusing root diameter with water uptake potential. In fact, larger roots are for carbohydrate storage and water transport, not uptake. The response appears to use diameter size to infer the uptake ability which isn't a great proxy.

*Authors: We thank the reviewer comment. We are aware of other root traits that contribute to uptake potential (diameter, length, SRL, etc.). We do not disregard measuring in the future any other trait that likely will result in important information. In this case, however we just focused on root diameter.*

**Comment 21.** Comment 7 – This response was confusing. Revising Figure 3 would be useful to clarify where the circular patterns and measurements are used would address this concern.

*Authors: Answer to this question was already addressed in comment 18.*

**Comment 22.** Comment 12 – I think it's important to mention you are using shielded antennas in the methods. While the majority of the antennas on the market fall into that category, it does make a difference in this methods paper as there are still a large number of non-shielded antennas in use.

*Authors: We thank the reviewer for this observation. We have emphasized the implementation of shielded antennas in this study throughout all the text in this manuscript.*

**Comment 23.** Comment 19 – this is the first mention of any understory vegetation. Could this provide any other error in the study?

*Authors: We are not sure from where this reference to the understory vegetation in the experimental plots was taken. In the response to comment 19 there was not any reference to understory vegetation (shown below). However, regarding the question we believe this factor is not a source of error in our system because understory vegetation is very low as a consequence of overgrazing by cattle and because roots of understory vegetation are much finer than the ones from the two forest species. Therefore, they are undetectable with the two antenna frequencies we used.*

Previous comment 19. Soils and root systems are highly variable and roots tend to crises cross/overlap areas in the soil, how did you partition roots out in the radar where they overlap (or grow side by side) and not consider them as 1 root, rather than 2 smaller roots). With such a low freq. antenna your diameters could be smaller roots in a group, rather than one 2.5 cm diameter root.

Authors: To solve the problem due to the hyperbolas overlapping as well to identify the correct position of roots, we applied a Stolt F-K Migration routine. With the application of the Stolt F-K Migration routine we were able to restore the real location and shape of reflecting boundaries in a section plane. On the other hand, to determine the wave propagation velocity, we relied on the "Hyperbola" tool (Rad Explorer software) in order to determine migration velocity.

Thus, after the Stolt F-K Migration routine was applied, if resulted reflections were not clear to interpret, that section was omitted.

**Comment 24.** Comment 21 – this response doesn't address the question. How did you detect anything under the rocks? If this site was really that dry, then the microsite question is less of a concern. But just lifting a rock to see if there's a root under it doesn't help you in detecting that specific root under a major reflector. The intent of the original comment wasn't about calibration, but rather how did you detect this data?

*Authors: The reviewer is correct, the GPR surveys were carried out during the dry ecohydrological period (Oct-Dec 2012) to minimize signal noise in the radargrams and facilitate root signal detection. Following a GPR survey and on-site interpretation of radargrams, we proceeded root validation through excavation, removing soil and removable rocks. The RAMAC XV Monitor for ProEx system, allowed us to visualize in situ, the position, size and depth of the root.*

[revised manuscript text omitted]